# BCDR: Betweenness Centrality-based Distance Resampling for Graph Shortest Distance Embedding

## Abstract

Along with unprecedented development in network analysis such as biomedical structure prediction and social relationship analysis, Shortest Distance Queries (SDQs) in graphs receive an increasing attention. Approximate algorithms of SDQs with reduced complexity are of vital importance to complex graph applications. Among different approaches, embedding-based distance prediction has made a breakthrough in both efficiency and accuracy, ascribing to the significant performance of Graph Representation Learning (GRL). Embedding-based distance prediction usually leverages truncated random walk followed by Pointwise Mutual Information (PMI)-based optimization to embed local structural features into a dense vector on each node and integrates with a subsequent predictor for global extraction of nodes' mutual shortest distance. It has several shortcomings. Random walk as an unstrained node sequence possesses a limited distance exploration, failing to take into account remote nodes under graph's shortest distance metric, while the PMI-based maximum likelihood optimization of node embeddings reflects excessively versatile local similarity, which incurs an adverse impact on the preservation of the exact shortest distance relation during the mapping from the original graph space to the embedded vector space.

To address these shortcomings, we propose in this paper a novel graph shortest distance embedding method called Betweenness Centrality-based Distance Resampling (BCDR). First, we prove in a statistical perspective that Betweenness Centrality(BC)-based random walk can occupy a wider distance range measured by the intrinsic metric in the graph domain due to its awareness of the path structure. Second, we perform Distance Resampling (DR) from original walk paths before maximum likelihood optimization instead of the PMI-based optimization and prove that this strategy preserves distance relation with respect to any calibrated node via steering optimization objective to reconstruct a global distance matrix. Our proposed method possesses a strong theoretical background and shows much better performance than existing methods when evaluated on a broad class of real-world graph datasets with large diameters in SDQ problems.

## 1 Introduction

Shortest Distance Queries (SDQs) in graphs focus on determining the minimum distance between arbitrary node pairs, defined as sum of related edge weights. Despite the fact that Shortest Path Queries (SPQs) (Wei, 2012; Sommer, 2014; Wang et al., 2021) are already renowned in graph structure exploration, SDQs play an essential role in increasing applications, such as social relationship analysis (Carlton, 2020; Melkonian et al., 2021; Chen et al., 2021; Zaki et al., 2021; Parveen & Varma, 2021), biomedical structure prediction (Yue et al., 2019; Galovičová et al., 2021; Sokolowski & Wasserman, 2021), learning theory (Yang et al., 2021; Yuan et al., 2021), optimization (Melkonian et al., 2021; Rahmad Syah et al., 2021; Jiang et al., 2021), etc. The key challenge in the SDQ problem is the prohibitive complexity in very large graphs. e.g., for an undirected dense graph with $N$ nodes and $k$ queries, the time complexity of A*(Hart et al., 1968) and Dijkstra Algorithm(Thorup & Zwick, 2004) are up to $O(kN^2)$ and $O(kN \log N)$ for unweighted and weighted graph, respectively.

Table 1: Overall comparison of approaches to shortest distance query.

Accuracy loss of approximate methods are evaluated on facebook(A.7.2) by mRE(2). $N$-number of nodes, $T$-walk length, $K$-window size, $l$-number of landmarks, $d$-dimension of embedding space, $\alpha, \beta, c$-constant independent of $N$.

| Objective | Method | Model | Off-Line Time | Space | Time | Accuracy Loss |
|---|---|---|---|---|---|---|
| Exact | Compress | - | $O(N^3)$ | $O(N^2)$ | $O(kN)$ | 0 |
| Exact | Index | - | $O(N^2 \log N)$ | $O(N^2)$ | $O(kN \log N)$ | 0 |
| Exact | Cache | - | $-$ | $O(N)$ | $O(N) \sim O(N^2)$ | 0 |
| Approximate | Oracle | - | $O(\alpha N^{1+\frac{1}{\alpha}})$ | $O(\alpha N^{1+\frac{1}{\alpha}})$ | $O(k\alpha)$ | $2(\alpha - 1)$ |
| Approximate | Landmark | - | $O(lN \log N + lN)$ | $O(lN)$ | $O(kl)$ | 0.523 |
| Approximate | Embedding | Orion | $O(\alpha N \log N + cN)$ | $O(dN)$ | $O(kd)$ | 0.688 |
| | | Rigel | $O(\alpha N \log N + cN)$ | $O(dN)$ | $O(kd)$ | 0.369 |
| | | DeepWalk | $O(TKN \log N + (T+c)N)$ | $O(dN)$ | $O(kd)$ | 0.329 |
| | | Node2Vec | $O(TKN \log N + (T+c)N)$ | $O(dN)$ | $O(kd)$ | 0.299 |
| | | DADL | $O(TKN \log N + (T+c)N)$ | $O(dN)$ | $O(kd)$ | 0.228 |
| | | BCDR(ours.) | $O(\alpha TKN \log N + (\beta T+c)N)$ | $O(dN)$ | $O(kd)$ | **0.177** |

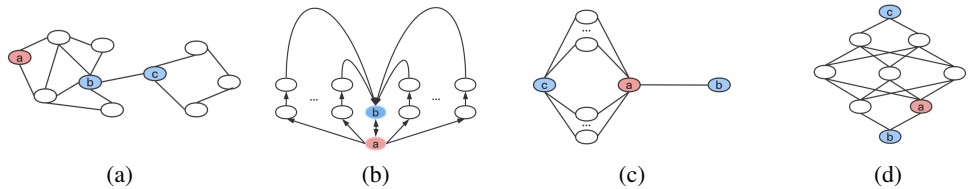

| (a) | (b) | (c) | (d) |

Figure 1: Distance confusion of a conventional model in embedding. **(a)** random walks rooted at $v_a$ have much difficulty in exploring beyond current community to $v_c$. **(b):** Weight decay on walks causes instability of shortest distance since walks rooted at $v_a$ have large probability to steer clear of $v_b$ for starters and back to $v_b$ as the end, which results in a extremely weak correlation between $v_a$ and $v_b$ despite the fact that they have an immediate edge. **(c):** A sufficient number of 2-hop links between $v_c$ and $v_a$ induce a shorter distance in embedding space than that of $v_b$ and $v_a$. **(d):** $v_b$ and $v_c$ sharing substantial connection deserve to be mapped closed to each other even if they have a large shortest distance gap, while the divergence of distance between $v_b, v_c$ and $v_a$ is also plagued with extraction.

To address this issue, a surging number of approximate algorithms in a wide range of fields have been proposed in the past a few years. They can be categorized into oracle-based (Thorup & Zwick, 2004; Baswana & Kavitha, 2006), landmark-based (Sarma et al., 2010; Gubichev et al., 2010), embedding-based distance prediction methods (Xiaohan et al., 2010; 2011; Rizi et al., 2018). Among these categories, embedding-based methods are of high accuracy with significantly reduced complexity (see Table1), owing much to rapid advances of representation learning in graphs (Perozzi et al., 2014; Grover & Leskovec, 2016). Embedding-based methods tackle the SDQ problem with two stages. First, they embed local structural features into a dense vector on each node, which preserves the essential information for indicating where the node is. Then, for arbitrary node pairs in the training set, subsequent predictor extracts mutual shortest distance globally via their embedding tuples and minimizes the mean of square loss between a predicted value and the real one. As predictors serve as a non-linear metric in the embedding space, the performance of embedding-based models depends highly on the first stage. These methods usually embed graph structures by leveraging truncated random walk to serialize node's neighborhood in a statistical perspective and maximize co-occurrence likelihood of nodes in one walk path to reflect their correlation, which is also proved to implicitly factorize a Pointwise Mutual Information (PMI) matrix (Levy & Goldberg, 2014; Shaosheng et al., 2015).

Although existing embedding-based methods have achieved a great success, they have several short-comings. On one side, random walk is an unstrained node sequence from the root, possessing a limited distance exploration. This is because each transition on nodes is not implied for a specific direction to move towards or beyond the root, especially after several walk steps, which is back-breaking to accommodate correlation with remote nodes under graph's shortest distance metric (see Figure 1a). On the other side, the PMI-based optimization reflects excessively versatile local sim-

ilarity, which is not guaranteed for shortest distance-preserved mapping from the original graph space to the embedded vector space. As a matter of fact, it exerts a too-general metric over nodes' correlation, wherein the more links or paths exist between two nodes, the stronger correlation they share. That means we have many ways to claim a shorter distance for two nodes (e.g., by adding mutual edges, deleting links to other nodes) even if some of the operations do not change their actual shortest path distance (see Figures 1c and 1d).

In this paper, we address the above shortcomings by proposing a novel graph shortest distance embedding method called Betweenness Centrality-based Distance Resampling (BCDR). Here, random walk paths are simulated by considering nodes' betweenness centrality on each transition for covering a wider distance range of nodes. Then, a sampling process of nodes based on their mutual shortest distances is performed before optimization of co-occurrence likelihood to preserve pairwise distance relation during the mapping from graph to the embedding space.

We summarize our major contributions as follows.

- We propose a **Betweenness Centrality (BC)-based random walk** for **accommodating correlation** with nodes **of wider distance range** under the intrinsic graph metric (see Section 3.1). To the best of our knowledge, there is no existing method that combines betweenness centrality and random walk for the graph representation. We prove in a statistical perspective that the transition driven by nodes' BC value tends to explore high order proximity of the root due to the awareness of local path structure.

- We propose a **Distance Resampling (DR)** strategy to **preserve** nodes' mutual **shortest distance during the mapping** from graph to the embedding space (see Section 3.2). We prove that by sampling node sequences from original walk paths before maximum likelihood optimization instead of PMI-based optimization, the objective can be steered to reconstruct a global distance matrix, and for any calibrated node, it strictly preserves the shortest distance relation to other nodes on the graph.

- We evaluate BCDR and compare it with existing embedding-based methods with a broad class of **real-world graph datasets with divergent diameters**. BCDR shows a **much better empirical performance** in solving the SDQ problem than existing embedding-based methods (see Section 4).

## 2 NOTATION & PROBLEM DEFINITION

### 2.1 NOTATION

$G = (V, E)$ denotes an undirected graph, with $V = \{v_1, v_2, ..., v_{|V|}\}$ being the set of nodes and $E$ being the set of edges. An edge $e_{ij} = (v_i, v_j)$ represents an undirected edge between nodes $v_i$ and $v_j$. A node $v_i$'s neighborhood $\mathcal{N}_i$ is a set of nodes with an edge with $v_i$, i.e., $\mathcal{N}_i = \{v_j | (v_i, v_j) \in E\}$. We use $Z_{|V| \times d}$ to represent an embedding matrix, where $d$ is the embedding size. For any matrix $B$, the symbol $\mathbf{B}_i$ represents the $i$-th row of $B$, and $B_{ij}$ represents the element at $i$-th row and $j$-th column. A truncated random walk $\mathcal{W}_i$ rooted at node $v_i$ of length $l$ is a random vector of $\langle \mathcal{W}_i^1, \mathcal{W}_i^2, \cdots, \mathcal{W}_i^l \rangle$, where $\mathcal{W}_i^k$ is a node chosen from the neighborhood of node $\mathcal{W}_i^{k-1}$ for $k = 1, ..., l$, with $\mathcal{W}_i^0 := v_i$. $P_i$ is a finite set of walk paths sampled from $\mathcal{W}_i$, and $W_i$ is the multiset of nodes on the walk paths in $P_i$.

### 2.2 PROBLEM DEFINITION

An arbitrary path $p$ of length $l \in \mathbb{N}$ on graph $G$ is an ordered sequence of nodes $(v_1, v_2, ...v_{l+1})$, where each node except the last one has an edge with the subsequent node, i.e., $(v_i, v_{i+1}) \in E$ for $1 \leq i \leq l$. The shortest path $\mathring{p}_{ij}$ is one of the paths with the minimum length $\mathring{l}$ between two nodes $v_i$ and $v_j$, with the shortest distance $D_{ij}$ defined as the length of $\mathring{p}_{ij}$. The global distance matrix $D$ comprises $\{D_{ij}\}$.

An embedding-based model learns two mappings $\mathcal{F}$ and $\mathcal{G}$ for structure embedding and mutual distance extraction, respectively, as follows.

$$\mathcal{F} : G \to \mathbb{R}^d, \quad \mathcal{G} : \langle \mathbb{R}^d, \mathbb{R}^d \rangle \to \mathbb{R} \tag{1}$$

A shortest distance query $Q$ can be simply defined as a set of node pairs, i.e., $Q = \{(v_{11}, v_{12}), (v_{21}, v_{22}), ..., (v_{|Q|,1}, v_{|Q|,2})\}$. For each node pair $(v_i, v_j)$, the model intends to find out approximation $\hat{D}_{ij} = \mathcal{G}(\mathcal{F}(v_i), \mathcal{F}(v_j))$ nearest to real $D_{ij}$. The commonly used metrics of approximation quality are mean of Relative Error (mRE) and mean of Absolute Error (mAE). mRE is defined as the relative loss of the prediction value with respect to the real value, while mAE measures the absolute gap between the prediction value and the real value, i.e.,

$$\text{mRE} := \frac{1}{|Q|} \sum_{(v_i, v_j) \in Q} \frac{|\hat{D}_{ij} - D_{ij}|}{D_{ij}}, \quad \text{mAE} := \frac{1}{|Q|} \sum_{(v_i, v_j) \in Q} |\hat{D}_{ij} - D_{ij}| \tag{2}$$

## 3 METHOD

Random walk as a serialization strategy of similarity measurement has been widely used in many graph representation and learning methods, aiming to model long-range dependency of nodes (Grover & Leskovec, 2016; Zhuang & Ma, 2018). The optimization of maximizing co-occurrence likelihood of nodes on walk paths also yields an impressive performance in network-structure extraction via approximating PMI matrix (Levy & Goldberg, 2014). But in terms of the shortest distance representation of a graph, we contend that this intuitive approach has several limitations on the performance. Consider a walk path $p = (v_a, v_{a_1}, v_{a_2}, \cdots, v_{a_l}) \in P_a$ sampled from a general random walk $\mathcal{W}_a$ from root node $v_a$. As an unstrained sequence of nodes, the distance measured along the walk path $p$ is not consistent with that on the graph (see Figure 1b), i.e., for $v_{a_i}, v_{a_j} \in p$,

$$i \leq j \not\Leftrightarrow D_{aa_i} \leq D_{aa_j} \tag{3}$$

where $i$ and $j$ are indexes of node $v_{a_i}$ and $v_{a_j}$ on $p$ and $1 \leq i, j \leq l$.

Using the aforementioned unstrained walk paths for maximizing co-occurrence likelihood incurs two problems.

1. **Problem 1: Limited exploration range of walks**. Each transition on walks only considers a local structure of the current node, causing agnostic tendency to move towards or beyond the root node under the graph's shortest-distance metric after a few steps (see Figure 1a).

2. **Problem 2: Intractability of shortest distance on paths**. Distance measured on walk paths may not actually reflect the graph's shortest distance because of the unbalanced number of links between different nodes (see Figure 1c and 1d).

In this section, we describe in detail our proposed method, which is a decent way of encoding shortest distances using BC-based random walk plus a distance resampling strategy, and present a theoretical analysis for its interpretability and efficiency. More specifically, we first present BC-based random walks and distance resampling, and then an algorithm to integrate them. We provide an intuitive complexity explanation. Additionally, we discuss the connection to Multidimensional Scaling (MDS) and graph structure decomposition in Appendix A.1 and A.2, respectively.

### 3.1 BC-BASED RANDOM WALK

**Definition 1.** *(Betweenness Centrality) Define $G = (V, E)$ as undirected graph. $v_i, v_s, v_t$ are arbitrary nodes in the node set $V$. $\sigma_{st}(v_i)$ represents the number of shortest paths between $v_s$ and $v_t$ that pass $v_i$, and $\sigma_{st}$ is the total number of shortest paths between $v_s$ and $v_t$. Then we say that BC of $v_i$ is*

$$\text{BC}(v_i) = \sum_{s \neq i \neq t} \frac{\sigma_{st}(v_i)}{\sigma_{st}} \tag{4}$$

To address Problem 1, we propose a BC-based random walk. As defined in Definition 1, $\text{BC}(v_i)$ determines the probability of $v_i$ located on shortest paths of arbitrary node pairs. Thus, we consider a node with large BC value vitally significant to drive the walk to move away from root node, since

it reveals a easy way of traveling to some nodes with minimal cost. And to leverage this property, in BC-based random walk $\mathcal{W}_a = \langle \mathcal{W}_a^1, \mathcal{W}_a^2, ... \mathcal{W}_a^j, ... \rangle$ on node $v_a$, we prefer choosing nodes with the largest BC values among their neighborhoods when simulating walk paths, i.e.,

$$p(\mathcal{W}_a^j | \mathcal{W}_a^{j-1} = v_{j-1}) = \frac{\text{BC}(\mathcal{W}_a^j)}{\sum_{v_k \in \mathcal{N}_{j-1}} \text{BC}(v_k)}, \ \mathcal{W}_a^j \in \mathcal{N}_{j-1} \tag{5}$$

where $\mathcal{N}_i$ is the neighborhood of $v_i$.

The following theorem, proved in Appendix A.3, indicates that a BC-based random walk tends to transit from $\mathcal{N}_a^{(h)}$ to $\mathcal{N}_a^{(h+1)}$, leading to a deeper exploration measured by the intrinsic graph's shortest distance. It also reveals that our method performs better when the number of final nodes $f_h(v_a)$ is larger or there are more links between final nodes $f_h(v_a)$ to connective nodes $e_h(v_a)$ at any $h$-order neighborhood of $v_a$.

**Theorem 1.** *Define $\mathcal{N}_a^{(h)} = \{v_j | D_{aj} = h\}$ as a set of nodes that are $h$-hops away from $v_a$ and $N_h(v_a) = |\mathcal{N}_a^{(h)}|$ as the number of nodes in the set. Let the number of the nodes that have connection with nodes in $\mathcal{N}_a^{(h+1)}$ (called connective nodes) be $e_h(v_a)$ and the number of other nodes (called final nodes) be $f_h(v_a)$. The BC we use is an approximate value by considering only the shortest path of nodes within a range of $k$-hops locally. Let $p_R(N_h(v_a) \to N_{h+1}(v_a))$ represent the probability to transit from nodes of $N_h(v_a)$ to $N_{h+1}(v_a)$ by a general random walk, and $p_B(N_h(v_a) \to N_{h+1}(v_a))$ represent that by a BC-based random walk. Let $P_R(f_h(v_a) \to e_h(v_a))$ be a transition from $f_h(v_a)$ nodes to $e_h(v_a)$. Then, for any node $v_a$ in graph $G$ and any $h > 1$,*

$$\frac{p_B(N_h(v_a) \to N_{h+1}(v_a))}{p_R(N_h(v_a) \to N_{h+1}(v_a))} = 1 + B(k) + C \tag{6}$$

$$\lim_{k \to \mathcal{E}(v_a) - 1 - h} B(k) + C = \frac{A_2 - 1}{A_1} + C > 0$$

$$A_1 = \frac{e_h(v_a)}{f_h(v_a)}, \quad A_2 = \frac{1}{p_R(f_h(v_a) \to e_h(v_a))}. \tag{7}$$

*where $C \geq 0$, and $\mathcal{E}(v_a)$ is the eccentricity of $v_a$, $\mathcal{E}(v_a) = \max_{v_b \in G} D_{ab}$.*

## 3.2 DISTANCE RESAMPLING

To address Problem 2, we propose a Distance Resampling (DR) strategy, which is inspired by Sampling-Importance-Resampling (SIR) method (Smith & Gelfand, 1992). Let $\mathbf{v}_x(v_1, v_2)$ be a random vector of the node tuple. Define $p(\mathbf{v}_x)$ as the joint distribution reflecting the real shortest distance of $v_1, v_2$ on the graph. We have

$$\max_f \mathbb{E}_{\mathbf{v}_x \sim p}[f] = \sum_{v_1, v_2 \in V} f(v_1, v_2) p(v_1, v_2) \tag{8}$$

as a probabilistic objective for representing the shortest distance $D_{12}$, where $f$ is a normalized learnable density function determined by nodes' mutual distance in the embedding space. According to the rearrangement inequality, the above equation arrives at the maximum when $f(v_1, v_2)$ varies consistent with $p(v_1, v_2)$, which means $f$ indicates the shortest distance metric on the graph with respect to any $v_1, v_2$ as $p$ does. Therefore, the critical issue is to extract accurate $p(v_1, v_2)$ in a graph. Let $q(\mathbf{v}_a)$ be the joint distribution reflecting distance relation on sampled walks. We try to leverage $q$ on walk paths to approximate $p$. Here, we adopt a simple but effective resampling by considering both their mutual distance $D_{12}$ as well as BC values.

Without loss of generality, let $v_i$ represent the root node of walks and $v_j$ be the $j$-th node on a walk path (i.e., sampled from $\mathcal{W}_i^j$). Our objective is thus to sample from $\mathcal{W}_i$ to generate a new random vector $\mathring{\mathcal{W}}_i = \langle \mathring{\mathcal{W}}_i^1, \mathring{\mathcal{W}}_i^2, ..., \mathring{\mathcal{W}}_i^{\mathring{l}} \rangle$ of length $\mathring{l}$. Then, each node $v_j$ sampled from $\mathring{\mathcal{W}}_i^j$ can be described as a weighted resampling in original walk paths $\mathcal{W}_i$ based on $D_{ij}$ and $\text{BC}(v_j)$, i.e.,

$$\mathbb{E}_{v_j \sim p(v_j | v_i)}[f | v_i] = \sum_{v_i, v_j \in V} f(v_i, v_j) p(v_j | v_i) = \sum_{v_i, v_j \in V} f(v_i, v_j) q(v_j | v_i) \frac{p(v_j | v_i)}{q(v_j | v_i)}$$

$$\approx \sum_{v_i \in V} \sum_{v_j \in W_i} f(v_i, v_j) q(v_j | v_i) \frac{\alpha^{D_{ij}} \text{BC}(v_j)}{\sum_{v_k \in W_i} \alpha^{D_{ik}} \text{BC}(v_k)} \tag{9}$$

where $\alpha$ is a hyper-parameter of the weight decay coefficient on paths.

Reminiscent of previous discussions on relating maximizing co-occurrence likelihood with matrix factorization, we demonstrate the connection between the above intuitive design and graph shortest distance metric. As we perform distance resampling for walk paths, the objective to learn an optimal node embedding is therefore interpreted as

$$
\begin{aligned}
\arg\max_{\mathbf{Z}} & \, \mathbb{E}_{\mathbf{v}_x \sim p}[f(\mathbf{Z}_i, \mathbf{Z}_j)] \\
& = \sum_{v_i \in V} p(v_i) \mathbb{E}_{v_j \sim p(\mathring{\mathcal{W}}_i)}[p(v_j|v_i)(\log \sigma(\mathbf{Z}_i \mathbf{Z}_j^T) + \lambda \mathbb{E}_{v_k \sim p_N(v_k|v_i)}[\log \sigma(-\mathbf{Z}_i \mathbf{Z}_k^T)])]
\end{aligned}
\tag{10}
$$

Thereinto, $f(\mathbf{Z}_i, \mathbf{Z}_j) = \log \sigma(\mathbf{Z}_i \mathbf{Z}_j^T)$, $p_N(v_k|v_i)$ is the distribution of negative sampling. In our algorithm, $p_N$ can be specified as a weighted random sampling over all occurred nodes in $W_i$ (simulated based on Equation 5) by occurrence frequency, i.e.,

$$
p_N(v_k|v_i) = \frac{\#(v_k \text{ occurs in } W_i)}{\#(W_i)}
\tag{11}
$$

where $\#(\cdot)$ is a counting function indicating the number of occurrence times of specified nodes, i.e., the cardinality of a sampled set.

**Proposition 1.** *Define $G = (V, E)$ as an undirected graph. $Z_{|V| \times d}$ is the embedding matrix of $V$ corresponding to maximizing likelihood objective $\mathbb{E}_{\mathbf{v}_x \sim p}[f(\mathbf{Z}_i, \mathbf{Z}_j)]$ defined in Equation 10. $p_N(v_b|v_a)$ is the negative sampling distribution of $v_b$ from $W_a$ simulated by Equation 11. Let the weight decay coefficient in distance resampling be $\alpha$, $0 < \alpha < 1$. Then, the inner product of embedding matrix corresponds to a global distance matrix, i.e. $ZZ^T = \hat{D}$. For any $v_a$ and any $v_b$ that is $n$-hops away from $v_a$, the distance between them in the embedding space varies linearly with respect to distance $n$, namely,*

$$
\hat{D}_{ab} = n \log \alpha - \log A
\tag{12}
$$

*where $A$ is a constant independent of $v_b$.*

Proposition 1 is proved in Appendix A.4.

Proposition 1 indicates that optimization of Equation 10 conforms to reconstruct a global distance matrix where nodes far away from each other in the graph under shortest distance metric(i.e., large $D_{ab}$) should be mapped with large distance in the embedding space(i.e., large $\hat{D}_{ab}$). We can also conclude that $\hat{D}_{ab}$ varies linearly with respect to the distance $n$ between two nodes, while $A$ is a constant independent of $v_b$ but related to $v_a$, which means when we fix the source (or destination) node as $v_a$, any destination (or source) node $v_b$'s distance with $v_a$ could be compared with each other (we call that distance is measurable with respect to calibrated node $v_a$).

Consider the preservation of shortest distance relation. Some studies on metric learning (Hermans et al., 2017; Zeng et al., 2020) have revealed that a tuple of samples $(v_a, v_b, v_c)$ being easy to learn means if $v_b$ shares strong correlation with $v_a$, the distance between $v_b$ and $v_a$ in the embedding space should be shorter than that of $v_c$ and $v_a$. With this property, we have the following theorem, which indicates that our method is consistent with distance relation under intrinsic graph metric. This will be used in the prediction task described next.

**Theorem 2.** *Each symbol here follows the definition in Proposition 1. Let $D$ be a global distance matrix defined on graph $G$ and $D_{ab}$ be graph's shortest distance between node $v_a$ and $v_b$. Then for any nodes $v_a, v_b, v_c \in G$,*

$$
(D_{ab} - D_{ac})(|\hat{D}_{ab}| - |\hat{D}_{ac}|) \geq 0
\tag{13}
$$

The proof is presented in Appendix A.5.

### 3.3 ALGORITHM

Our BCDR algorithm is presented in Algorithm 1. It contains three steps. First, pre-computation of the BC value (line 21) is required for each node on the graph. Second, for each node $v_i$, we sample a batch of walk paths $P_i$ guided by the BC value according to Section 3.1 (line 1 to 20). Meanwhile,

Table 2: Statistics of graph datasets and their train & validation & test sets.

RoBC-range of BC, mBC-mean of BC, $L$-landmark nodes, $\mathcal{T}$-train set, $\mathcal{V}$-validation set, $\mathcal{E}$-test set.

| | $|V|$ | $|E|$ | $|E|/|V|$ | Diameter | RoBC | mBC | $|L|$ | $|\mathcal{T}|$ | $|\mathcal{V}|$ | $|\mathcal{E}|$ |
|---|---|---|---|---|---|---|---|---|---|---|
| Cora | $2,708$ | $10,787$ | $3.9834$ | $21$ | $375.20$ | $2.7174$ | $100$ | $100 \times |V|$ | $800 \times 30$ | $1,600 \times 10$ |
| Facebook | $4,039$ | $176,437$ | $21.846$ | $8$ | $1,306.9$ | $0.89931$ | $100$ | $100 \times |V|$ | $800 \times 50$ | $1,600 \times 20$ |
| GrQc | $5,242$ | $30,042$ | $5.7310$ | $17$ | $148.43$ | $2.7219$ | $100$ | $200 \times |V|$ | $800 \times 70$ | $1,600 \times 30$ |

distance relation of each node with $v_i$ is also recorded for subsequent resampling. Third, for each node $v_i$, we resample nodes in $W_i$ by their distances with $v_i$ and BC values to formulate $\mathring{P}_i$ (line 25 to 34), which is optimized by maximizing pair-wise co-occurrence likelihood (line 37). This embedding algorithm takes $O(w_{out}l_{out}^2|V|\log|V| + (w_{in}l_{in} + w_{out}l_{out})|V|)$ of time complexity and $O((w_{in}l_{in} + w_{out}l_{out})|V|)$ of space complexity for any sparse graph (detailed analysis is presented in Appendix A.6). The above result reveals a feasible way of reducing time complexity by just taking larger $w_{in}l_{in}$ and smaller $w_{out}l_{out}$ in BCDR. This practice is applied in our experiments.

---

**Algorithm 1:** BCDR Embedding Algorithm

**Input:** input graph $G = (V, E)$, embedding size $d$, input sample size $w_{in}$, output sample size $w_{out}$, input walk length $l_{in}$, output walk length $l_{out}$, distance weight decay $\alpha$, use smoothed normalization $\tau$.

1 **Def** BC_Walk $(G, v_i, D_i, w_{in}, l_{in}, \tau)$ :
2    **for** *walk $k$ from 0 to $w_{in}$* **do**
3      visit sign set $S_i := \{v_i\}$, current node $v_c := v_i$, length $c_i := 0$, real length $c_i^r := 0$
4      **while** $c_i < l_{in}$ **do**
5        $\mathcal{N}_c^k := \{v_j | v_j \in \mathcal{N}_c \wedge v_j \notin S_i\}$
6        **if** $\tau$ *is True* **then**
7          normalize $p_{c\to j} \leftarrow \text{softmax}(\gamma_j + 1), v_j \in \mathcal{N}_c^k$
8        **else**
9          normalize $p_{c\to j} \leftarrow \frac{\gamma_j}{\sum_{v_m \in \mathcal{N}_c^k} \gamma_m}, v_j \in \mathcal{N}_i^k$
10      **end**
11      sample next $v_n$ from $\mathcal{N}_c^k$ by $p_{c\to j}$
12      **if** $v_n \in D_i$ **then**
13        $D_i[v_n] = \min\{D_i[v_n], c_i^r + 1\}$
14        $c_i^r = \min\{D_i[v_n] - 1, c_i^r\}$
15      **else**
16        $D_i[v_n] = c_i^r + 1$
17      **end**
18      $v_c \leftarrow v_n, S_i \leftarrow v_n, c_i \leftarrow c_i + 1, c_i^r \leftarrow c_i^r + 1$
19    **end**
20 **end**

21 pre-compute BC of each node $v_i$ as $\gamma_i \leftarrow \text{BC}(v_i)$.
22 Walk path set $P$
23 **for** $v_i$ *in $G$* **do**
24    distance map $D_i := \{v_i : 0\}$
25    $D_i \leftarrow$ BC_Walk $(G, v_i, D_i, w_{in}, l_{in}, \tau)$
26    **for** $v_j \in D_i.keys$ **do**
27      **if** $\tau$ *is True* **then**
28        $\gamma_j' \leftarrow \text{softmax}(\gamma_j + 1)$
29        $p(v_j | v_i) := \alpha^{D_i[v_j]} \cdot \gamma_j'$
30      **else**
31        $p(v_j | v_i) := \alpha^{D_i[v_j]} \cdot \gamma_j$
32      **end**
33    **end**
34    sample walk paths $P_i = \{(v_i, v_{x_1}, v_{x_2}, \cdots, v_{x_j}, \cdots)\}$ of length $l_{out}$ by $p(v_{x_j}|v_i), v_{x_j} \in D_i.keys$ for $w_{out}$ times.
35    append $P_i$ into $P$
36 **end**
37 maximize Equation 10 by $P$.
38 **return** Z

---

## 4 EXPERIMENTAL EVALUATION

In this section, we evaluate our model's performance with several graph datasets and compare it with conventional embedding-based methods for SDQ tasks.

### 4.1 DATASETS

We test our model using three real-world graph datasets as well as some simulated datasets. As this paper focuses on the shortest distance prediction of connected and undirected graphs, we make a trivial change for some directed graphs by simply eliminating the direction of each edge. For graphs with multiple connected components, we repeatedly add one edge between two unconnected components until the whole graph is connected. The graphs we use are of complex inner connections and divergent diameters. The basic information of these datasets is outlined in Table 2 (columns 2 to 7). The detailed information is presented in Appendix A.7.

We define $\mathcal{T}$, $\mathcal{V}$, and $\mathcal{E}$ as datasets for train, validation, and test, respectively. Each of them is composed of $(v_i, v_j, D_{ij})$. As simulations of the above sets in huge graphs have to be of $O(|V|)$ or less complexity, in our experiments, we initially select a group $\mathcal{S}$ of nodes as sources, and for each source $v_i \in \mathcal{S}$, a destination set $\mathcal{D}$ with their shortest distances to the source is randomly collected. Then, $|\mathcal{S}| \times |\mathcal{D}_i|$ input node pairs are simulated. The detailed setup for each dataset is presented in Table 2 (columns 8 to 11).

### 4.2 PARAMETER SETUP

We compare our method with conventional graph embedding models based on random walk as well as matrix factorization (see Appendix A.8). We simulate $40$ walks on each node for random walk-based methods, and each walk is truncated at a length of $40$. Sliding window size and negative sampling size are fixed at $20$ and $5$, respectively, for each dataset. Node2Vec takes hyper-parameter

Table 3: Performance comparison of embedding-based models

| | Cora | | Facebook | | GrQc | |
|---|---|---|---|---|---|---|
| | mAE | mRE | mAE | mRE | mAE | mRE |
| LLE | $5.6265 \pm 0.0490$ | $0.8445 \pm 0.0096$ | $1.9921 \pm 0.0731$ | $0.6841 \pm 0.0029$ | $4.8849 \pm 0.1034$ | $0.7105 \pm 0.0165$ |
| GF | $5.6249 \pm 0.0876$ | $0.8440 \pm 0.0142$ | $1.8743 \pm 0.0717$ | $0.6383 \pm 0.0284$ | $4.8562 \pm 0.0516$ | $0.7125 \pm 0.0084$ |
| LE | $5.6393 \pm 0.0782$ | $0.8455 \pm 0.0132$ | $2.0312 \pm 0.0625$ | $0.6998 \pm 0.0248$ | $5.0046 \pm 0.0242$ | $0.7366 \pm 0.0044$ |
| DeepWalk | $1.5183 \pm 0.0654$ | $0.2425 \pm 0.0101$ | $0.9323 \pm 0.0272$ | $0.3289 \pm 0.0137$ | $2.8002 \pm 0.1479$ | $0.4169 \pm 0.0227$ |
| Node2Vec | $1.3072 \pm 0.0236$ | $0.2115 \pm 0.0038$ | $0.8541 \pm 0.0436$ | $0.2993 \pm 0.0166$ | $1.5156 \pm 0.0691$ | $0.2278 \pm 0.0087$ |
| DADL | $1.1349 \pm 0.0180$ | $0.1790 \pm 0.0031$ | $0.6150 \pm 0.0325$ | $0.2279 \pm 0.0135$ | $1.3624 \pm 0.1366$ | $0.2033 \pm 0.0183$ |
| BCDR | $\mathbf{0.9768 \pm 0.0245}$ | $\mathbf{0.1605 \pm 0.0043}$ | $\mathbf{0.4804 \pm 0.0406}$ | $\mathbf{0.1770 \pm 0.0156}$ | $\mathbf{1.0490 \pm 0.0634}$ | $\mathbf{0.1684 \pm 0.0058}$ |

$p = q = 1$. Regularization $r$ of GF is set to $1.0$. DADL takes the Hadamard operator for embedding aggregation. Like previous research (Zhuang & Ma, 2018), linear regression is utilized as a predictor for general graph embedding models.

For our model, we take the same configuration as previous random walk-based models using input walk length $l_{in} = 40$ and input sample size $w_{in} = 40$. In order to keep complexity of parameters equal to or even less than that of baselines, the multiplication of output length $l_{out}$ and output sample size $w_{out}$ is fixed at 40, with $l_{out} = 10, w_{out} = 160$ for Cora and GrQc and $l_{out} = 40, w_{out} = 40$ for Facebook. Meanwhile, the weight decay coefficient $\alpha$ is set to $0.1$ for Cora and GrQc datasets and $0.98$ for Facebook. To avoid dramatically unbalanced values of BC, we also take smoothed normalization ($\tau = True$) in all datasets except Facebook. Finally, all of the above models use the Adam optimizer with a learning rate $\epsilon_r = 1e - 4$ for training, and embedding dimension $d$ is fixed at 16 for every dataset.

### 4.3 PERFORMANCE OF PREDICTION ACCURACY

To compare embedding-based methods, we train each model on each dataset up to 500 iterations and save the results every 10 epochs. The best models are selected by their mAE and mRE scores on validation set $\mathcal{V}$. Each model is evaluated 5 times independently on each dataset, and their average performance is recorded. The resulting mAE and mRE are reported in Table 3. Some comparisons with the latest GRL methods and further discussion are presented in Appendix A.9, and run-time test compared with general random walk is reported in Appendix A.10. We can see from the table that our model outperforms previous models significantly for all three datasets.

### 4.4 RESULT OF EXPLORATION DISTANCE

As stated in Section 3.1, exploration distance under intrinsic graph metric plays an indispensable role in modeling a wider range of node correlation. Here, we compare our BC-based random walk with existing renowned walk strategies, including DeepWalk (Perozzi et al., 2014; Zhuang & Ma, 2018), Node2Vec (Grover & Leskovec, 2016), and Random Surfing (Cao et al., 2016). We use $TG(20, 1, 3, 10)$ as a test graph of large diameters. We randomly sample 20 root nodes and, for each root, simulate 10 walks with length 10 to show how many nodes in different order proximity are visited. The ideal situation for a batch of walk paths with length $l$ is to cover up to nodes $l - hop$ away from the current root. The results are shown in Figure 2. From the figure, we can see that our BC-based walk is much more competitive regarding exploration distance. A further illustration of diverse graph structures is presented in Appendix A.11.

### 4.5 PRESERVATION OF DISTANCE RELATION

We have also evaluated the property of shortest distance preservation during the mapping from the original graph space into embedded vector space. First, we test distance variation in the embedding space when adjusting the shortest distance between nodes on the graph, taking the same configuration of simulation dataset as $TG(20, 1, 3, 10)$. Distance in the embedding space is measured by inner product $\mathbf{Z}_i\mathbf{Z}_j^T$ for any node $v_i$ and $v_j$. We initially train embedding vectors using walks simulated by each model and randomly sample 20 source nodes with 100 destinations for each source. The results are shown in Figure 3, which indicates that our model has a better tendency to maintain a linear distance relation for the mapping.

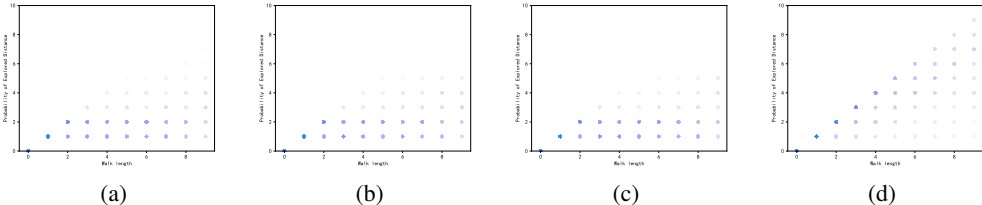

Figure 2: Exploration distance of different random walk strategies tested on graph $\mathrm{TG}(20, 1, 3, 10)$. **(a):** general random walk in DeepWalk. **(b):** Node2Vec. **(c):** Random Surfing. **(d):** BC-based random walk(ours.).

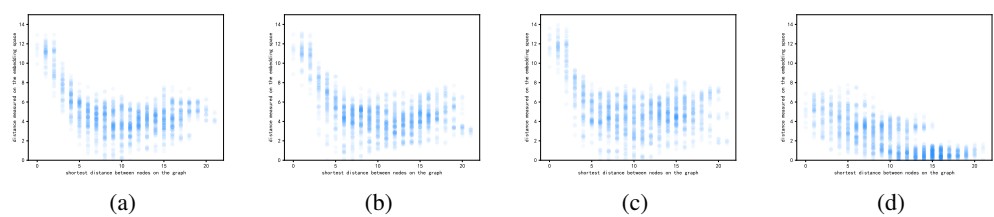

Figure 3: Distance relation during mappings when taking different random walk strategies. **(a):** general random walk in DeepWalk. **(b):** Node2Vec. **(c):** Random Surfing. **(d):** BCDR(ours.).

Second, we try to find out how much the probability distance relation is violated in the embedding space. We randomly take 10000 node triple $(v_a, v_b, v_c)$, and record if they violate Equation 13. The results are shown in Figure 4. The figure confirms that our model is much more satisfactory in preserving distance relation during mappings.

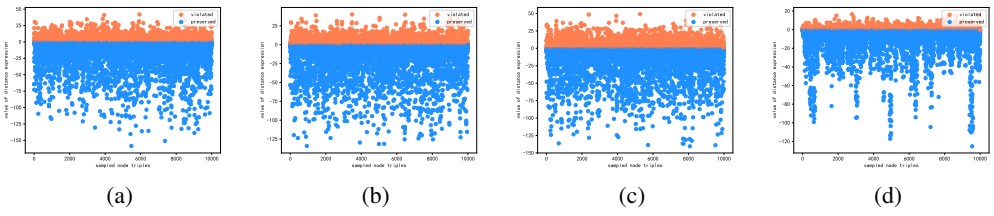

Figure 4: Distance preservation in the embedding spaces of different models. **(a):** general random walk in DeepWalk. **(b):** Node2Vec. **(c):** Random Surfing. **(d):** BCDR(ours.).

## 5 CONCLUSION

In this paper, we propose a novel graph shortest distance embedding method called Betweenness Centrality-based Distance Resampling (BCDR). It improves the graph embedding for the shortest distance representation with two components we propose in this paper. The first is Betweenness Centrality-based random walk to accommodate long-distance correlation on graphs by covering a wider range of nodes under the intrinsic graph metric. The second is a distance resampling strategy to preserve shortest distances during the mapping from graph to the embedding space via reconstructing a global distance matrix. The experimental evaluation indicates that BCDR possesses a better capacity than existing graph embedding methods to extract distance structure from original graphs. BCDR can be integrated into graph-based learning models (especially in graph neural networks), which should improve their performance on graph structure recognition. This will be our future work.

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

# A APPENDIX

## A.1 CONNECTIONS TO MDS

Our algorithm also shares some connection with conventional MDS methods in global distance metric. Take a view of original MDS. Input data $X$ distributes in agnostic high-dimensional euclidean space and mutual distance between samples is extractable as a global distance matrix $D^X$. A approximate distance matrix $D^Y$ calculated by embedding matrix $Y$ endeavor to be optimized closest to $D^X$ by Frobenius norm, i.e.,

$$\min_Y \|D^X - D^Y\|_F^2 = \|D^X - YY^T\|_F^2 \tag{14}$$

Here, we suppose $D^X$ and $Y$ have been double-centered for stability, and finally, $Y$ could be deduced as the group of top-d leading eigenvectors in $D^X$. Then, as a non-linear version of MDS, Isomap (Tenenbaum et al., 2000) generalizes $D^X$ as distribution on a manifold and utilizes short hops to measure the mutual distance between samples. Deriving from the above idea, we regard the global distance matrix $D^X$ as a canonical metric on graphs and exert embedding matrix $Z_{|V|\times d}$ to reconstruct it. Different from previous work, we encounter prohibitive complexity to acquire all elements in $D^X$ and leverage an iterative process to approximately minimize the accuracy loss between $D^Z$ and $D^X$.

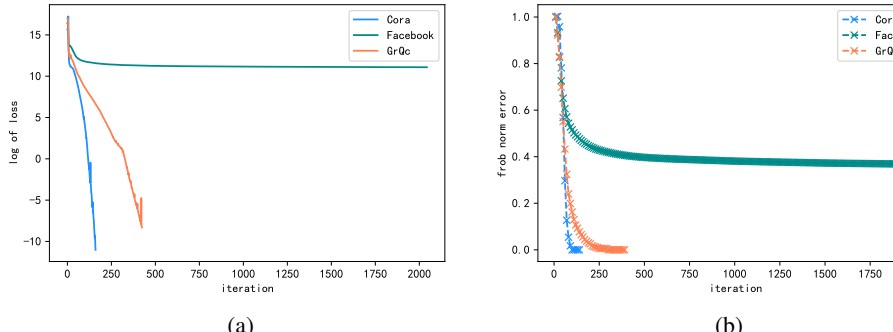

(a)                                                       (b)

Figure 5: Converge analysis of LPCA on different datasets **(a):** LPCA loss. **(b):** Frobenius norm error.

Table 4: Converge time and Accuracy Comparison between LPCA and BCDR(ours.).

|  | Cora | | Facebook | | GrQc | |
|---|---|---|---|---|---|---|
|  | Time | mRE | Time | mRE | Time | mRE |
| LPCA (Chanpuriya et al., 2020) | **82.60s** | 0.3445 | $2587s$ | 0.8272 | $1163s$ | 0.3952 |
| BCDR(ours.) | $113.7s$ | **0.1576** | **774.0s** | **0.1599** | **264.8s** | **0.1603** |

## A.2 CONNECTIONS TO GRAPH STRUCTURE DECOMPOSITION

A newly published work (Chanpuriya et al., 2020) provides an exciting perspective to embed complex sparse graphs perfectly into low-rank representations, which is helpful for downstream ML tasks. Nevertheless, we need to address the limitation of LPCA in shortest distance prediction as follows.

First and most importantly, it should be clarified that a perfect representation of graph structure is not equal to that of graph shortest path structure, since they have a large calculation gap. According to Floyd-Warshall algorithm (Floyd, 1962), even if each node is aware of all related path structures, inference of the exact shortest path structure also needs up to $O(N^3)$ complexity. Second, LPCA has poor embedding performance on some relatively dense graphs, despite converging fast on sparse graphs. We test this method on the three real-world datasets used in this paper, and illustrate the converge curve in Figure 5. The converge time is reported in Table 4. The results show LPCA meets a bottleneck on the relative-dense graph (Facebook) and consumes a long time to converge.

Compared to the above method, the motivation of this paper is to directly embed graph shortest distance matrix with sub-linear time complexity for fast and accurate online queries of shortest distance. To the best of our knowledge, there is no existing embedding method that could directly and perfectly represent shortest path structures in linear time. Moreover, our method takes a more flexible objective to represent shortest distances, since only high-level restrictions are implicitly exerted on the embedding space by Equation 12 and 13.

## A.3 PROOF OF THEOREM 1

*Proof.* We simplify symbols $N_h(v_a), e_h(v_a), f_h(v_a)$ as $N_h, e_h, f_h$ for short.

$$\overrightarrow{N}_h = \#(\bigcup_{i=h}^{\min\{\mathcal{E}(v_a), h+k-1\}} \{v_j | v_j \in \mathcal{N}_a^{(i)}\})$$
$$\overleftarrow{N}_h = \#(\bigcup_{i=\max\{0, h-k+1\}}^{h} \{v_j | v_j \in \mathcal{N}_a^{(i)}\})$$
(15)

According to definition in the theorem, we have $N_h = e_h + f_h$. Since only $e_h$ nodes could travel from $N_h$ to $N_{h+1}$, We firstly consider $P_R(e_h \rightarrow e_{h+1}|e_h \rightarrow N_{h+1})$ and $P_B(e_h \rightarrow e_{h+1}|e_h \rightarrow N_{h+1})$.

For general random walk, the choice of destination is based on uniform sampling, thus causing

$$P_R(e_h \rightarrow e_{h+1}|e_h \rightarrow N_{h+1}) = \frac{e_{h+1}}{N_{h+1}} \tag{16}$$

For BC-based random walk, we need calculate BC value of $e_{h+1}$ and $f_{h+1}$ nodes for starters. Let $\text{BC}(e_{h+1})$ and $\text{BC}(f_{h+1})$ represent the BC value of nodes in $e_{h+1}$ and $f_{h+1}$ respectively, and the correspond legal shortest path counts comes from 4 sources as $\{\overleftarrow{N}_h \rightarrow \overrightarrow{N}_{h+2}\}$, $\{\overleftarrow{N}_h \rightarrow N_{h+1}\}$, $\{N_{h+1} \rightarrow \overrightarrow{N}_{h+2}\}$ and $\{N_{h+1} \rightarrow N_{h+1}\}$. And we use $\text{BC}(\{\cdot \rightarrow \cdot\})$ as the BC gain from the specified source, then

$$
\begin{aligned}
\text{BC}(\{\overleftarrow{N}_h \rightarrow \overrightarrow{N}_{h+2}\}) &= \overleftarrow{N}_h \overrightarrow{N}_{h+2} \\
\text{BC}(\{\overleftarrow{N}_h \rightarrow N_{h+1}\}) &= \overleftarrow{N}_h(f_{h+1} \cdot 0 + e_{h+1}\beta_e^{(1)}) \\
\text{BC}(\{N_{h+1} \rightarrow \overrightarrow{N}_{h+2}\}) &= \overrightarrow{N}_{h+2}(f_{h+1} \cdot 1 + e_{h+1}\beta_e^{(1)}) \\
\text{BC}(\{N_{h+1} \rightarrow N_{h+1}\}) &= N_{h+1}^2 \beta_e^{(2)}
\end{aligned}
\tag{17}
$$

where $\beta_e^{(1)}$ means average BC gain between nodes in $e_{h+1}$ and $\overrightarrow{N}_{h+2}$, and $\beta_e^{(2)}$ means average BC gain between nodes both in $e_{h+1}$, which are constantly related with $G$. Therefore, we have

$$\text{BC}(e_{h+1}) = \overleftarrow{N}_h \overrightarrow{N}_{h+2} + (\overleftarrow{N}_h + \overrightarrow{N}_{h+2})e_{h+1}\beta_e^{(1)} + \overrightarrow{N}_{h+2}f_{h+1} + N_{h+1}^2\beta_e^{(2)} \tag{18}$$

Likewise, we calculate

$$
\begin{aligned}
\text{BC}(f_{h+1}) &= \overleftarrow{N}_h \overrightarrow{N}_{h+2} \cdot 0 + \overleftarrow{N}_h(f_{h+1}\beta_f^{(1)} + e_{h+1} \cdot 0) + \overrightarrow{N}_{h+2}(f\beta_f^{(1)} + e_{h+1} \cdot 0) + N_{h+1}^2\beta_e^{(2)} \\
&= (\overleftarrow{N}_h + \overrightarrow{N}_{h+2})\beta_f^{(1)} + N_{h+1}^2\beta_f^{(2)}
\end{aligned}
\tag{19}
$$

To compare the above $\text{BC}(e_{h+1})$ and $\text{BC}(f_{h+1})$,

$$\frac{\text{BC}(f_{h+1})}{\text{BC}(e_{h+1})} = \frac{(\overleftarrow{N}_h + \overrightarrow{N}_{h+2})\beta_f^{(1)} + N_{h+1}^2\beta_f^{(2)}}{\overleftarrow{N}_h \overrightarrow{N}_{h+2} + (\overleftarrow{N}_h + \overrightarrow{N}_{h+2})e_{h+1}\beta_e^{(1)} + \overrightarrow{N}_{h+2}f_{h+1} + N_{h+1}^2\beta_e^{(2)}} \tag{20}$$

note that for any $N_j$ and $N_{(x,y)} = \overrightarrow{N}_0 - \overleftarrow{N}_x - \overrightarrow{N}_y$ where $j, x, y \in \{0, \mathcal{E}(v_a)\}$,

$$\lim_{k \rightarrow \mathcal{E}(v_a)-1-h} \frac{N_j}{N_{(x,y)}} = \lim_{k \rightarrow \mathcal{E}(v_a)-1-h} \epsilon(k) = 0 \tag{21}$$

Equation 19 is reduced to

$$\frac{\text{BC}(f_{h+1})}{\text{BC}(e_{h+1})} = \frac{2\epsilon(k)\beta_f^{(1)} + \epsilon(k^2)\beta_f^{(2)}}{1 + 2\epsilon(k)ne_{h+1}\beta_e^{(1)} + \epsilon(k)f_{h+1} + \epsilon(k^2)\beta_e^{(2)}} = 2\beta_f^{(1)}\epsilon(k) \tag{22}$$

Then, we perform weighted random sampling based on BC and get

$$p_B(e_h \rightarrow e_{h+1}|e_h \rightarrow N_{h+1}) = \frac{e_{h+1}}{e_{h+1} + 2f_{h+1}\beta_f^{(1)}\epsilon(k)} \tag{23}$$

Now, we consider the relation between $P_R(N_h \to e_{h+1}|N_h \to N_{h+1})$ and $P_B(N_h \to e_{h+1}|N_h \to N_{h+1})$.

$$\frac{p_B(N_h \to N_{h+1})}{p_R(N_h \to N_{h+1})} = \frac{p_B(e_h)p_B(e_h \to N_{h+1})}{p_R(e_h)p_R(e_h \to N_{h+1})}$$

$$= \frac{p_B(N_{h-1} \to e_h) + p_B(N_{h-1} \to f_h)p_B(f_h \to e_h)}{p_R(N_{h-1} \to e_h) + p_R(N_{h-1} \to f_h)p_R(f_h \to e_h)}$$

$$= 1 + \frac{[p_B(N_{h-1} \to e_h) - p_R(N_{h-1} \to e_h)][1 - p_R(f_h \to e_h)]}{p_R(N_{h-1} \to e_h) + p_R(N_{h-1} \to f_h)p_R(f_h \to e_h)}$$

$$+ \frac{p_R(f_h \to e_h)\epsilon}{p_R(N_{h-1} \to e_h) + p_R(N_{h-1} \to f_h)p_R(f_h \to e_h)} \quad (24)$$

$$= 1 + \frac{f_h[1 - 2f_h\beta_f^{(1)}\epsilon(k)][1 - p_R(f_h \to e_h)]}{(1 + \frac{f_h}{e_h})[e_h + 2f_h\beta_f^{(1)}\epsilon(k)]p_R(f_h \to e_h)}$$

$$+ \frac{p_R(f_h \to e_h)\epsilon}{p_R(N_{h-1} \to e_h) + p_R(N_{h-1} \to f_h)p_R(f_h \to e_h)}$$

Let $C = \frac{p_R(f_h \to e_h)\epsilon}{p_R(N_{h-1} \to e_h) + p_R(N_{h-1} \to f_h)p_R(f_h \to e_h)}$, $B(k) = \frac{f_h[1 - 2f_h\beta_f^{(1)}\epsilon(k)][1 - p_R(f_h \to e_h)]}{(1 + \frac{f_h}{e_h})[e_h + 2f_h\beta_f^{(1)}\epsilon(k)]p_R(f_h \to e_h)}$. Since $C$ is a non-negative value independent of $k$, finally we get

$$\frac{p_B(N_h \to N_{h+1})}{p_R(N_h \to N_{h+1})} = 1 + B(k) + C \quad (25)$$

where

$$\lim_{k \to \mathcal{E}(v_a)-1-h} B(k) + C = \frac{A_2 - 1}{A_1} + C. \quad (26)$$

$\square$

### A.4 PROOF OF PROPOSITION 1

*Proof.* Initially, we rewrite the negative sampling item of Equation 10 as

$$\mathbb{E}_{v_k \sim p_N(v_k|v_i)}[\log \sigma(-\mathbf{Z}_i\mathbf{Z}_k^T)]) = \sum_{v_k \in W_i} p_N(v_k|v_i)[\log \sigma(-\mathbf{Z}_i\mathbf{Z}_k^T)]$$

$$= p_N(v_j|v_i)[\log \sigma(-\mathbf{Z}_i\mathbf{Z}_j^T)] + \sum_{v_k \in W_i \setminus \{v_j\}} p_N(v_k|v_i)[\log \sigma(-\mathbf{Z}_i\mathbf{Z}_k^T)] \quad (27)$$

Then, for each pair of $v_i \in V$ and $v_j \in W_i$, we get independent objective by combing similar items in overall likelihood expression $\mathbb{E}_{\mathbf{v}_x \sim p}[f(\mathbf{Z}_i, \mathbf{Z}_j)]$, and get

$$\mathbb{E}_{\mathbf{v}_x \sim p}[f(\mathbf{Z}_i, \mathbf{Z}_j)] = \sum_{v_i \in V} \sum_{v_j \in W_i} \mathcal{L}(v_i, v_j) \quad (28)$$

$$\mathcal{L}(v_i, v_j) = p(v_i, v_j) \log \sigma(-\mathbf{Z}_i\mathbf{Z}_j^T) + \lambda p(v_i)p_N(v_j|v_i) \log \sigma(-\mathbf{Z}_i\mathbf{Z}_j^T)$$

Let $\hat{D} = ZZ^T$, for each node pair $(v_a, v_b)$ with mutual shortest distance $\hat{D}_{ab}$, consider

$$\hat{D}_{ab} = \mathbf{Z}_a\mathbf{Z}_b^T = \arg\max_{\mathbf{Z}_a, \mathbf{Z}_b} \mathcal{L}(v_a, v_b) \quad (29)$$

Suppose $v_b$ is $n$-hops away from $v_a$, and denote $BC(v_b)$ by $\gamma_b$, according to Equation 5,

$$p(v_a, v_b) = p(v_a) \cdot p(v_b|v_a) = p(v_a) \cdot \alpha^n \gamma_b \quad (30)$$

Note that $p_N(v_b|v_a)$ is a negative sampling in walk set $W_a$(Equation 11) which is selected by BC-based random walk locally restricted to $v_a$ according to Equation 5, we have

$$p_N(v_b|v_a) = \kappa(v_a)\gamma_b \quad (31)$$

where $\kappa(v_a)$ is related with the neighbor structure of $v_a$. Then, Equation 29 could be described as

$$\hat{D}_{ab} = \arg\max_{\hat{D}_{ab}} p(v_a)\alpha^n \gamma_b \log \sigma(\hat{D}_{ab}) + \lambda p(v_a)\kappa(v_a)\gamma_b \log \sigma(-\hat{D}_{ab}) \tag{32}$$

Solve the above problem by just let $\frac{\partial \mathcal{L}(v_a, v_b)}{\partial \hat{D}_{ab}}$ be equal to zero, i.e.,

$$\frac{\partial \mathcal{L}(v_a, v_b)}{\partial \hat{D}_{ab}} = p(v_a)\alpha^n \gamma_b \sigma(1 - \hat{D}_{ab}) - \lambda p(v_a)\kappa(v_a)\gamma_b \sigma(1 + \hat{D}_{ab}) = 0 \tag{33}$$

After some simplification, we get

$$\hat{D}_{ab} = n \log \alpha - \log \lambda \kappa(v_a) \tag{34}$$

Let $A = -\lambda\kappa(v_a)$ and there holds

$$\hat{D}_{ab} = n \log \alpha - \log A \tag{35}$$

$\square$

## A.5 PROOF OF THEOREM 2

*Proof.* Let $v_b$ and $v_c$ in graph be $n$ and $m$-hops away from $v_a$ respectively. In terms of node pair $(v_a, v_b)$, as proved in Proposition 1, their mutual distance $D_{ab}$ in the embedding space varies linear with respect to the graph shortest distance $n$, i.e.,

$$|\hat{D}_{ab}| = |n \log \alpha - \log A| = -n \log \alpha + \log A \tag{36}$$

Likewise, we have for $(v_a, v_c)$

$$|\hat{D}_{ac}| = |m \log \alpha - \log A| = -m \log \alpha + \log A \tag{37}$$

where $0 < \alpha < 1$ and $A$ is independent of $v_b$ and $v_c$. Then, consider the distance relation of $v_a$, $v_b$ and $v_c$, there holds

$$(D_{ab} - D_{bc})(|\hat{D}_{ab}| - |\hat{D}_{bc}|) = (n - m)(m - n)\log \alpha = -\log \alpha \cdot (n - m)^2 \geq 0 \tag{38}$$

$\square$

## A.6 COMPLEXITY ANALYSIS OF BCDR EMBEDDING ALGORITHM

We analyze the complexity of Algorithm 1 as follows.

The first step depends on the algorithm used for BC calculation, wherein some approximation methods could reduce complexity to $O(|V|)$ or $O(|V|\log|V|)$(Brandes & Pich, 2007; Ercsey-Ravasz & Toroczkai, 2010). Then, in the second step, for each walk rooted at each node $v_i$(line 23), we use a loop(line 2 to 20) up to $w_{in}$ times for generating nodes on the walk. Normalization of $p_{i \rightarrow j}$(line 5 to 18) is calculated for $l_{in}$ times for each node, and summation of $\gamma$ is up to $O(|E|)$ during the whole routine. Therefore, it takes totally $O(w_{in}l_{in}|V| + |E|)$ time complexity. Finally, as for the third step, line 26 to 33 re-weights explored area of every node $v_i$ which need loops up to $w_{in}l_{in}|V|$ times. For the sake of resampling $w_{out}l_{out}$ nodes, line 34 also requires normalization by $O(w_{out}l_{out}|V|)$. Line 37 use a maximum likelihood optimization which occupies $O(w_{out}l_{out}^2|V|\log|V| + w_{out}l_{out}|V| + |E|)$ time complexity(Mikolov et al., 2013a). As for space complexity, distance map $D_i$ and $p(v_j|v_i)$ are stored temporarily up to $O(w_{in}l_{in})$, while visit sign set $S_i$ is up to $O(w_{in})$. For dumped and resampled walk paths, $O((w_{in}l_{in} + w_{out}l_{out})|V|)$ space is required. Finally, our embedding algorithm takes $O(w_{out}l_{out}^2|V|\log|V| + (w_{in}l_{in} + w_{out}l_{out})|V| + |E|)$ time complexity and $O((w_{in}l_{in} + w_{out}l_{out})|V|)$ space complexity. Especially, for sparse graph, time complexity is reduced to $O(w_{out}l_{out}^2|V|\log|V| + (w_{in}l_{in} + w_{out}l_{out})|V|)$.

## A.7 COMPLEMENTARY INFORMATION OF DATASETS

This part provides detailed information about graph datasets we used, and presents some visualization result in Figure 6.

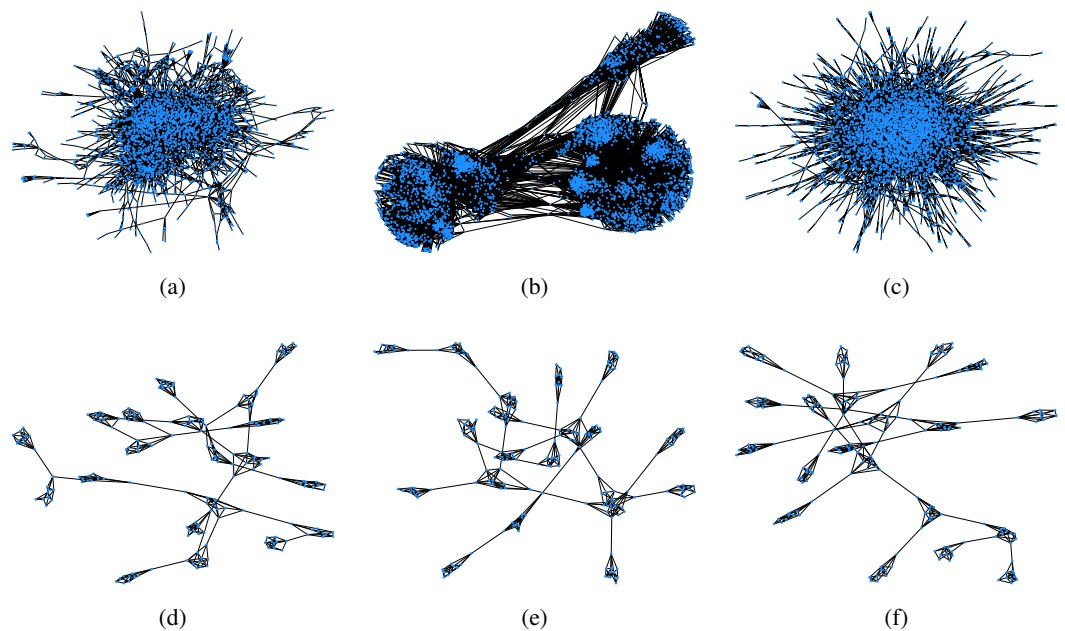

Figure 6: Visualization of graph datasets. **(a):** Cora. **(b):** Facebook. **(c):** GrQc. **(d), (e), (f):** TG$(20, 1, 3, 10)$.

### A.7.1 CORA

Cora graph dataset describes the citation relationship of papers, which contains 2708 nodes and 10556 directed edges among them. Each node also has a predefined feature with 1433 dimensions.

### A.7.2 FACEBOOK

Facebook dataset(Leskovec & Krevl, 2014) describes the relationship among Facebook users by their social circles(or friend lists), which is collected from a group of test users. Facebook has also encoded each user with a reindexed user ID to protect their privacy.

### A.7.3 GRQC

Arxiv GR-QC (General Relativity and Quantum Cosmology) collaboration network(Leskovec & Krevl, 2014) is recorded from the e-print arXiv in the period from January 1993 to April 2003, which used to represent co-author relationship based on their submission. We suppose an undirected edge $(v_i, v_j)$ if an author $v_i$ co-authored a paper with another author $v_j$. If one paper is owned by $k$ authors, a complete graph of $k$ nodes is generated correspondingly.

### A.7.4 TEST GRAPH (TG)

We define TG$(cls, c, r, n)$ as a parameterized graph simulation for SDQ tasks. $cls$ is the number of sub-graphs to represent communities, and $n$ is the number of nodes in each community. $c$ and $r$ determine the inner and outer connectivity of each community, respectively. TG is guaranteed to be an undirected and connected graph in our experiment, which is also expected to be sparse and of large diameter. Some simulation results of TG$(20, 1, 3, 10)$ that we used in Section 4.4 and 4.5 are presented in Figure 6 (d), (e), (f).

### A.7.5 OTHER GRAPHS

We also use some other graphs with diverse structures in this paper. The visualization of these graphs is shown in Figure 7. We describe each graph as follows.

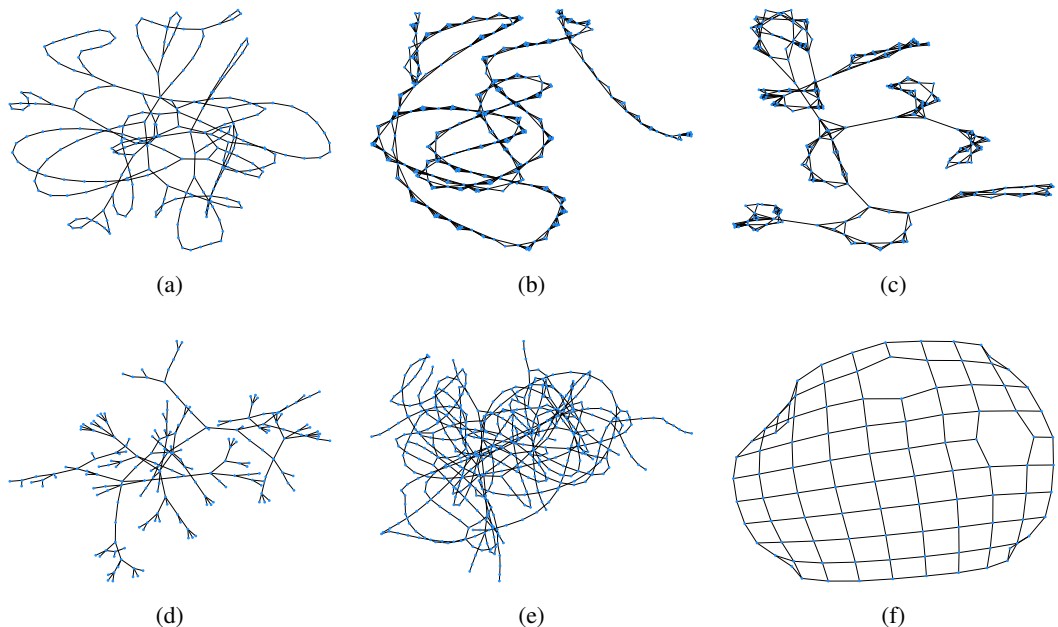

Figure 7: Visualization of some synthetic graphs with diverse structure. **(a):** circle graph. **(b):** triangle graph. **(c):** tri-circle graph. **(d):** tree graph. **(e):** spiral graph. **(f):** net graph.

- Circle Graph: a graph that contains several circles of different sizes. The simulation of circle graphs takes an iterative process where for each newly introduced circle, there are a limited number of nodes (called exit nodes) connected to the previous node set.

- Triangle Graph: a graph possessing several cliques which are linearly connected mutually.

- Tri-circle Graph: a graph that combines the properties of circle graphs and triangle graphs. Here, each circle is simulated by connecting triangle sub-graphs end to end.

- Tree Graph: a graph that is generated from one root to several leaves recursively. There is no cycle in tree graphs. To control the tree structure, we define a splitting probability that is decayed exponentially with current depth.

- Spiral Graph: a graph shaped like a spiral line. We first simulate a line graph and add edges between nodes with exponentially increased distances by their indices on the line.

- Net Graph: a graph containing grid-like connections between nodes. We define a small probability to drop those edges stochastically.

## A.8 BASELINE ALGORITHM

We outline all of the baseline models used in performance comparison as follows.

### A.8.1 LLE

Locally Linear Embedding(LLE)(Roweis & Saul, 2000) is an effective method of dimensional reduction that preserves the local linear combination property of nodes. In contrast with previous PCA(Jol, 2002) and LDA(Blei et al., 2003) which are guaranteed to discover an optimal reduction in euclidean space, LLE that addresses local distance relation gives a more compact representation for non-linear manifolds.

### A.8.2 LE

Analog to Laplace-Beltrami operator on manifolds, Laplacian Eigenmap(LE)(Roweis & Saul, 2000) leverages graph Laplacian matrix to generate embedding vectors. The objective of LE is to minimize

Table 5: Performance comparison with other GRL techniques

| | Cora | | Facebook | | GrQc | |
|---|---|---|---|---|---|---|
| | mAE | mRE | mAE | mRE | mAE | mRE |
| GraRep (Shaosheng et al., 2015) | 2.4036 | 0.3444 | 2.7975 | 1.0000 | 4.2263 | 0.6083 |
| NetMF (Qiu et al., 2018) | 4.1388 | 0.5988 | 1.5179 | 0.5477 | 4.10134 | 0.6037 |
| VERSE (Tsitsulin et al., 2018) | 2.9453 | 0.4140 | 1.1553 | 0.3939 | 3.3925 | 0.4761 |
| LPCA (Chanpuriya et al., 2020) | 2.3739 | 0.3445 | 2.0306 | 0.8272 | 2.7971 | 0.3952 |
| BCDR(ours.) | $0.9768 \pm 0.0245$ | $0.1605 \pm 0.0043$ | $0.4804 \pm 0.0406$ | $0.1770 \pm 0.0156$ | $1.0490 \pm 0.0634$ | $0.1684 \pm 0.0058$ |

mutual distance on edges by $L_2$ loss, which maps nodes with more first-order links closer to each other.

### A.8.3 GF

Graph Factorization(GF)(Ahmed et al., 2013) utilize matrix factorization methods(Gemulla et al., 2011) to deal with graph structure exploration. By approximating the graph adjacency matrix on a bundle of random nodes, GF is of high efficiency for large graphs.

### A.8.4 DEEPWALK

DeepWalk(Perozzi et al., 2014) is a random walk-based embedding method of sub-linear complexity to represent graph structure. This method derives from skip-gram(Mikolov et al., 2013a) and negative sampling(Mikolov et al., 2013b) to learn each node representation by predicting its context.

### A.8.5 NODE2VEC

Node2Vec(Grover & Leskovec, 2016) improves DeepWalk by performing a parametrized random walk. For each transition, Node2Vec considers second-order proximity to decide the next node be near or far from the previous node, which allows exploration both in local and global scopes.

### A.8.6 DADL

DADL(Rizi et al., 2018) leverages conventional GRL methods to embed local structure on each node, which implies some similarities between the current node and its neighbors. Then, by considering different binary operations over node embeddings in deep learning techniques, this method outperforms conventional ones in the shortest distance prediction task.

### A.9 FURTHER COMPARISON IN ACCURACY

We also evaluate our embedding method with the latest GRL techniques. Parameter setup takes the same configuration as stated in Section 4.2 except for some trivial modification to match the embedding dimension $d = 16$. All compared models take default parameters. We list out the comparison among these methods in Table 5. Moreover, we further illustrate the relation between prediction accuracy and path length for each method on both sparse graph (GrQc) and relatively-dense graph (Facebook). The sampling frequency of different path lengths in each dataset is presented in Figure 8. Length-level prediction accuracy is presented in Figure 9.

The experimental results show our method outperforms others mainly benefited by decent representations of node pair's similarity in a large range of shortest distance. On the one hand, BC-based walk occupies a wider range of shortest paths on each node, making remote nodes available before embedding. On the other hand, the distance resampling strategy discerns nodes by different order proximity to preserve their mutual shortest distance.

### A.10 RUN TIME OF BCDR AND GENERAL RANDOM WALK

As is stated in Section 4.2, the time complexity of BCDR could be adjusted by sampling parameters $w_{in}, w_{out}, l_{in}, l_{out}$. To keep the capacity of accomodating large shortest distance, $w_{in}$ and $l_{in}$ are fixed as the previous. Define $\beta$ as the compressing coefficient of path length, and use $l_{out} = 40 \times \beta$. $w_{out}$ is thus adapted for inputs, i.e., $w_{out} = \frac{l_{in}w_{in}}{l_{out}} = \frac{40}{\beta}$. The comparasion between BCDR

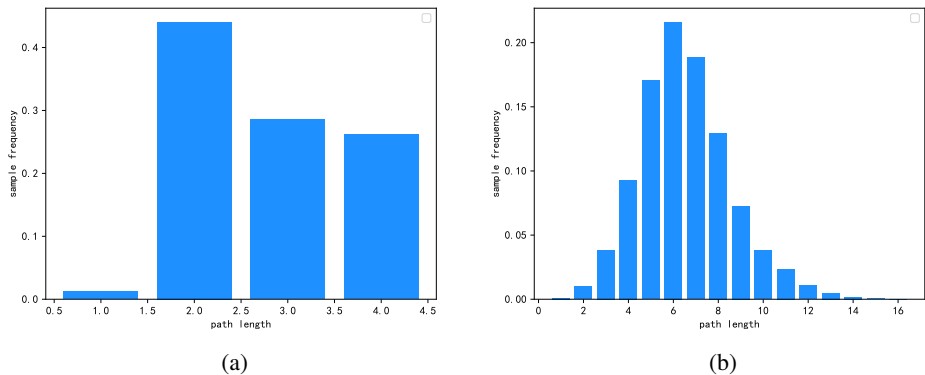

Figure 8: Sampling frequency of different path lengths. **(a):** Facebook dataset. **(b):** GrQc dataset.

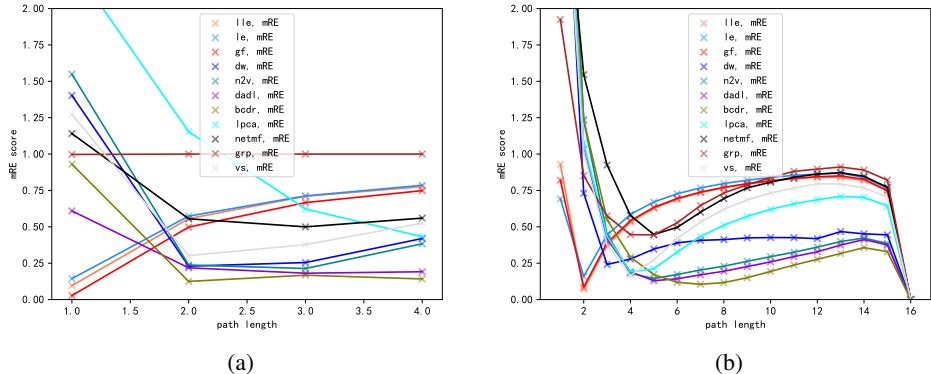

Figure 9: Length-level prediction accuracy **(a):** Facebook dataset. **(b):** GrQc dataset.

and general random walk method is provided in Figure 10 and 11 for run time and corresponding accuracy, respectively.

The experimental results reveal that our method could reduce time complexity with high accuracy to keep pace with general random walk in sparse graphs, but for relatively-dense graphs(like Facebook), there is some room for further optimization.

### A.11 FURTHER DISCUSSION ON WALKING PATTERNS

We test the performance of BC-based walk and other walking patterns on six synthetic graphs. The statistics of these graphs we simulated are listed in Table 6. To address the aspect of getting out of local cliques or circles, we provide another BFS-like searching pattern where each transition tends to choose the edges that could get out of the local clique by considering up to second-order proximity (with probability $p_{out} \approx 0.901$). Properties and visualization of these graphs are presented in Appendix A.7.5. And traversal results are illustrated in Figure 12 to Figure 17. The fluctuation of BC value on each graph is illustrated in Figure 18

The results are analyzed as follows.

- For circle graphs and tri-circle graphs, BC-based walk tends to choose the exit nodes of each circle since they share a large BC gain by locating on the shortest path between inner nodes and outer nodes.

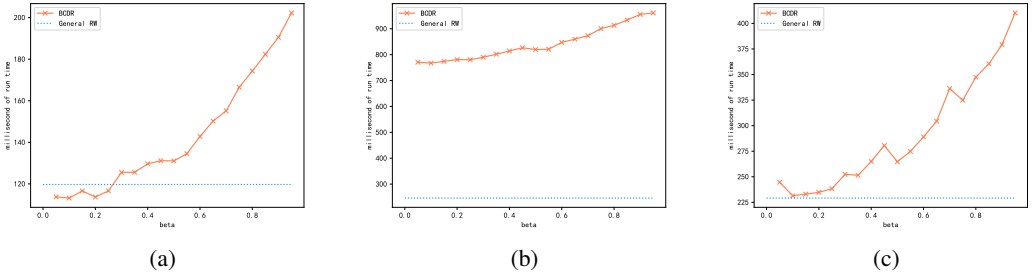

Figure 10: Comparison of run time between BCDR and general random walk. **(a):** Cora dataset **(b):** Facebook dataset. **(c):** GrQc dataset.

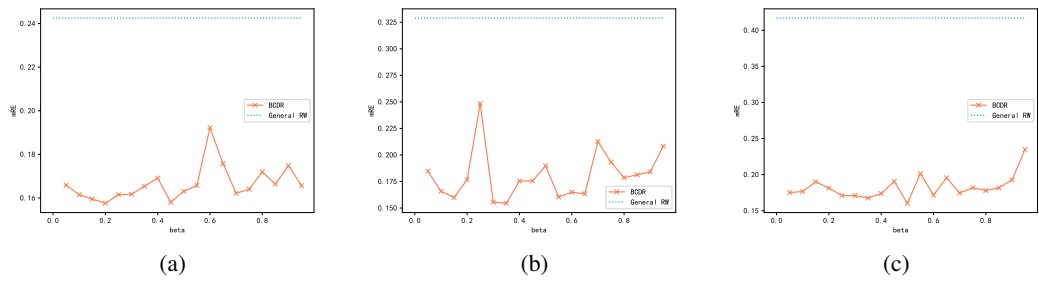

Figure 11: Comparison of accuracy correpsonding to run time presented in Figure 10. **(a):** Cora dataset **(b):** Facebook dataset. **(c):** GrQc dataset.

- For triangle graphs, transitions on every triangle clique tend to move forward since the number of nodes beyond the current clique is often larger than that of inner nodes, contributing to more shortest paths.

- For tree graphs, each transition appears to move forward from the root to the leaves, and all walking patterns show a similar exploration.

- For spiral graphs, although some exit nodes are located on the circle, BC gains on exit nodes and inner nodes on circles are usually on par, which misleads the direction choice of the next transition. Thus, BC-based walk only performs slightly better than others.

- For net graphs, all walking patterns are limited on exploration distance, since BC values on different nodes do not change considerably.

To conclude, the above results show that BC-based walk possesses a better all-around performance to capture remote nodes with large shortest distances.

Table 6: Statistics of six synthetic graphs

|  | Circle | Triangle | Tri-circle | Tree | Spiral | Net |
|---|---|---|---|---|---|---|
| $|V|$ | 262 | 352 | 234 | 207 | 550 | 100 |
| $|E|$ | 281 | 853 | 559 | 206 | 585 | 178 |
| $|E|/|V|$ | 1.0725 | 2.4233 | 2.3889 | 0.9952 | 1.0636 | 1.78 |
| avg. Degree | 2.1412 | 4.8438 | 4.7735 | 1.9855 | 2.1255 | 3.55 |
| diameter | 32 | 100 | 26 | 18 | 101 | 18 |

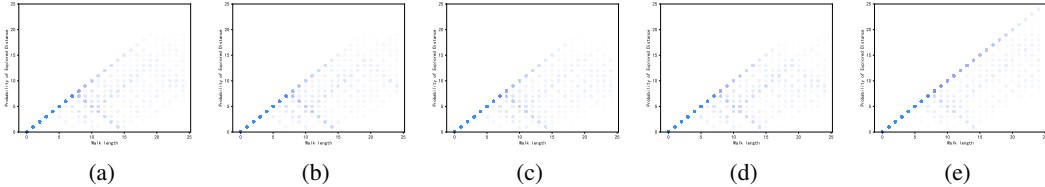

Figure 12: Exploration distance of different random walk strategies tested on circle graph. **(a):** General random walk **(b):** Node2Vec. **(c):** Random Surfing. **(d):** BFS-like search. **(e):** BC-based random walk(ours.).

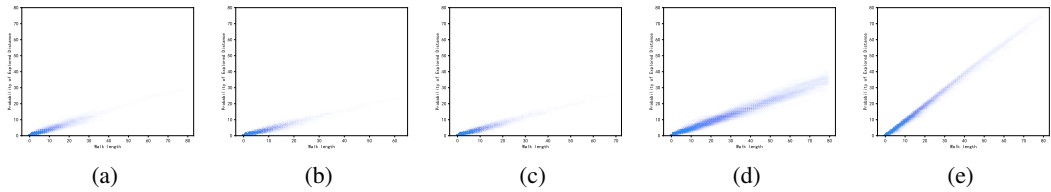

Figure 13: Exploration distance of different random walk strategies tested on triangle graph. **(a):** General random walk **(b):** Node2Vec. **(c):** Random Surfing. **(d):** BFS-like search. **(e):** BC-based random walk(ours.).

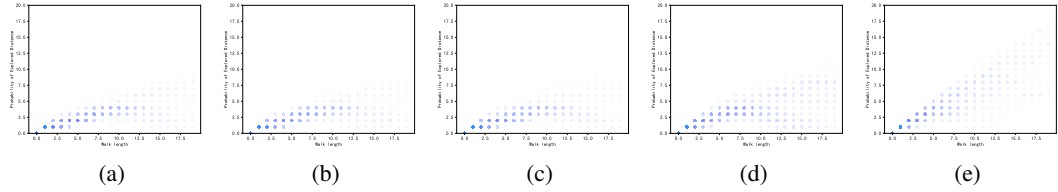

Figure 14: Exploration distance of different random walk strategies tested on tri-circle graph. **(a):** General random walk **(b):** Node2Vec. **(c):** Random Surfing. **(d):** BFS-like search. **(e):** BC-based random walk(ours.).

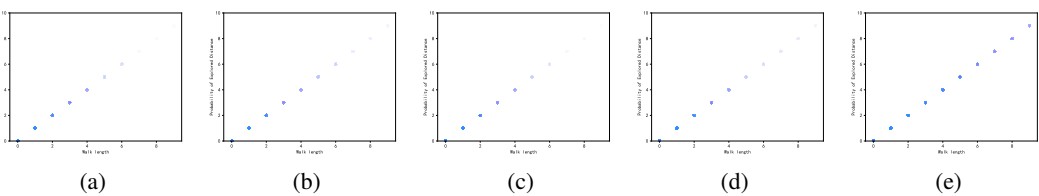

Figure 15: Exploration distance of different random walk strategies tested on tree graph. **(a):** General random walk **(b):** Node2Vec. **(c):** Random Surfing. **(d):** BFS-like search. **(e):** BC-based random walk(ours.).

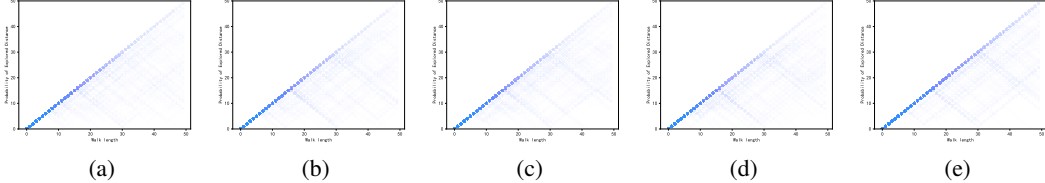

Figure 16: Exploration distance of different random walk strategies tested on spiral graph. **(a):** General random walk **(b):** Node2Vec. **(c):** Random Surfing. **(d):** BFS-like search. **(e):** BC-based random walk(ours.).

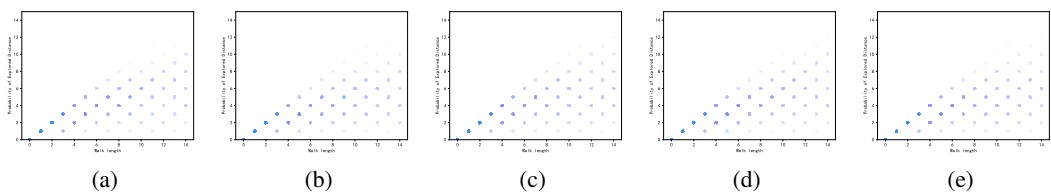

Figure 17: Exploration distance of different random walk strategies tested on net graph. **(a):** General random walk **(b):** Node2Vec. **(c):** Random Surfing. **(d):** BFS-like search. **(e):** BC-based random walk(ours.).

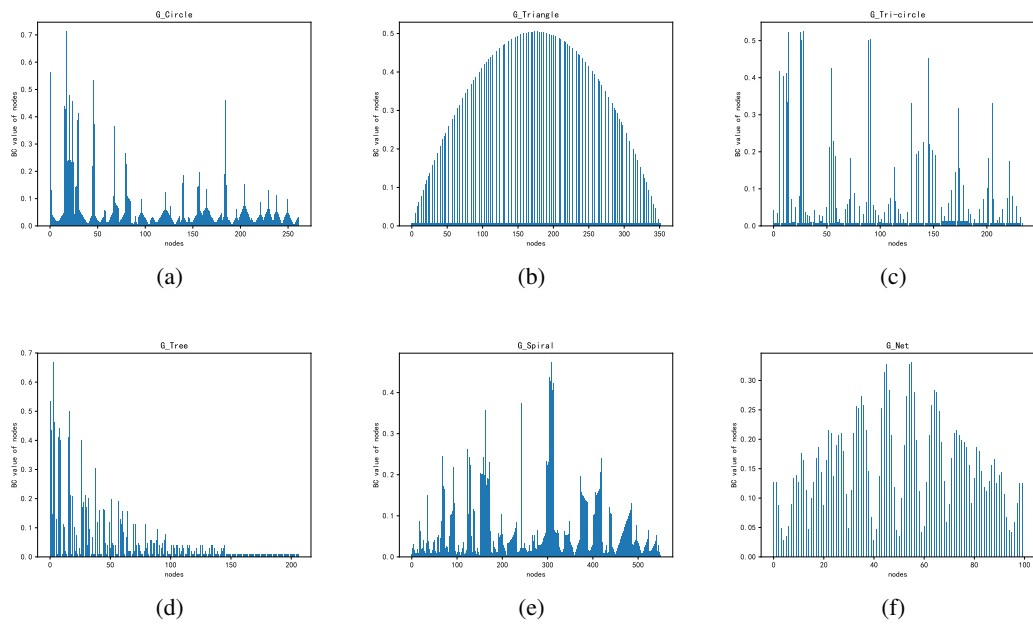

Figure 18: Fluctuation of BC value on each graph. **(a):** circle graph. **(b):** triangle graph. **(c):** tri-circle graph. **(d):** tree graph. **(e):** spiral graph. **(f):** net graph.

