# OpenReview forum: "BCDR: Betweenness Centrality-based Distance Resampling for Graph Shortest Distance Embedding"
_ICLR.cc/2022/Conference — ICLR 2022 Submitted_

### Official Review · Reviewer_UCXV · 2021-11-01

**Correctness:** 1
**Technical Novelty And Significance:** 3
**Empirical Novelty And Significance:** 3
**Recommendation:** 6
**Confidence:** 4

**Details Of Ethics Concerns:**

The work is mostly algorithmic and does not have direct impact on biased solutions in its current form.

**Main Review:**

Among the strengths, we can mention a couple. Using the betweenness centrality to guide the random walk to scape local constraints of the truncated random walker is intriguing and based on the evidence provided it seems promising.
The method considers not only ensuring the random walk takes into account longer distance associations among nodes but also the representation of shortest distances.
With respect to the drawbacks, random walks, even when betweenness is consider, can be affected by cycles in the graph structure. How is this handle by the technique is not thoroughly described in the article.
Also, while the paper is relatively well organized, there are many details that are defer to the appendix, including the proofs of the main theorems. There should be at least a sketch of the proof in the main body of the paper to have a self-contain text.
Finally, random walk betweenness centrality is a hot topic of research and is relevant and complementary to the main idea of the paper. Thus, works in this area should be cited.

**Summary Of The Paper:**

This paper presents a new Shortest Distance Query technique, namely, the Betweenness Centrality-based Distance Re- sampling (BCDR). The objective of this technique is overcome some drawbacks of traditional embedding-based distance prediction methods based on truncated random walks and point-wise Mutual Information (PMI). These drawbacks are a limited distance exploration and the lack of preservation of the shortest distance relation due to local optima. BCDR uses betweenness centrality to create a random walk that occupies a wider distance and uses Distance Resampling (DR) instead of PMI to preserve the relationship of distances. The paper presents theoretical guarantees of performance to address the exploration range and the intractability of shortest distance on paths. The experiments on three real datasets and simulated datasets show the performance evaluation with respect to baselines, exploration distance, and preservation of distance relation (and violation of the probability distance relation)

**Summary Of The Review:**

Overall, the paper has strengths as I highlighted above that merit publication as long as the clarifications on the policy for dealing with cycles and the clarifications on the proofs is provided. I recommend also including some related work on betweenness centrality for random walks and random walk betweenness centrality.

---

> ### Author Response · Authors · 2021-11-16
> **Response to Reviewer UCXV**
>
> > Random walks, even when betweenness is consider, can be affected by cycles in the graph structure. How is this handle by the technique is not thoroughly described in the article.
>
> We admit cycles in the graph may preclude random walk strategies from performing BFS-like traversal on graph structures. But it should be noted that cycles only induce **minimal impact** on BC-based walks. The reasons are listed as follows.
>
> - Theoretically, each transition in the BC-based walk depends on the **BC value** of the next nodes. BC value of inner nodes in a circle is **only** gained from the inner shortest paths. Comparatively, exit nodes of the current circle gain BC value **between inner and outer** nodes. Since the number of outer nodes is often **considerable** (if not, an elaborate search in the clique would be better),  our method tends to choose the exit nodes as the next with a **large probability**, which promises a **wider** traversal range.
> - Experimentally, we further evaluate our model on some synthetic graphs with **different cycle structures** (including circle, triangle, tri-circle, etc.). The properties and visualization of these graphs are available in Appendix A.7.5. Some further comparisons of different walking patterns in Appendix A.11 show our method has obvious advantages to **traverse beyond circles** since it is benefited a lot from the fluctuation of BC value on different nodes (see Fig. 18).
>
> > While the paper is relatively well organized, there are many details that are defer to the Appendix, including the proofs of the main theorems.
>
> We admit sufficient information should have been provided in the main text, despite some details being omitted due to the limitation of paper length.
>
> > Random walk betweenness centrality is a hot topic of research and is relevant and complementary to the main idea of the paper.
>
> We admit related work about random walk strategy and betweenness centrality should be included, and will append them later in this paper.

---

> > ### Comment · Reviewer_UCXV · 2021-11-23
> > **Thank you and suggestion**
> >
> > Thank you very much for your reply. It could be great if you could add the information you mention in your answer from a quantitative performance perspective to the final version of your paper. Particularly, an evaluation of how the density of cycles on more central nodes vs. peripheral nodes affect the performance of your approach can be useful as the that density changes could facilitate understanding the strengths and weaknesses of your approach.

---

> > > ### Author Response · Authors · 2021-11-24
> > > **Response to Reviewer UCXV**
> > >
> > > Thank you for your valuable suggestions. We are willing to provide some quantitative evaluations regarding cycles for both density and resident node types. Further discussion will be added to the final paper soon.

---

### Official Review · Reviewer_7mP1 · 2021-11-01

**Correctness:** 4
**Technical Novelty And Significance:** 3
**Empirical Novelty And Significance:** 3
**Recommendation:** 6
**Confidence:** 2

**Main Review:**

Strength: This paper proposed a novel approach for graph shortest distance embedding. The betweenness centrality based random walk and distance resampling techniques are interesting, which more efficiently explore the graph and preserve the shortest distance information. They also provide theoretical analysis and justification for this method. The experimental results also show the effectiveness of this new algorithm.

Weakness: The notations in section 3 are slightly complicated. Especially, some notations are used and discussed in high-level ideas without explanation and definition.

Questions:
1. In proposition 1, should the embedding distance matrix take an opposite sign, since the current linear expression is negative?
2. For proper parameter alpha and sufficiently large distance pair vi and vj, does proposition 1 imply a constant distance distortion for this embedding?


After responses:
Thank other reviewers for their comments and discussions. Also, the authors' responses clearly answered my questions. I think this paper provided some interesting ideas. But, I agree with other reviewers that this paper can be improved by polishing the statement of claims and including those valuable discussions and comparisons. Especially, those comparisons with LPCA and combinatorial approaches.


**Summary Of The Paper:**

This paper proposed a new graph shortest distance embedding method. This method uses a betweenness centrality based random walk to sample paths in the graph and distance resampling step before optimization. They show that the estimated distance after embedding has a linear dependence with the original distance in the graph They also show that this embedding method preserves the shortest distance relation between points. In experimental results, they show that this algorithm achieves better accuracy than previous algorithms.

**Summary Of The Review:**

The method proposed in this paper is interesting. Two techniques used in this new algorithms and analysis of this algorithm are non-trivial. This algorithm also achieves a good performance in experiments.

---

> ### Author Response · Authors · 2021-11-16
> **Response to Reviewer 7mP1**
>
> > In proposition 1, should the embedding distance matrix take an opposite sign, since the current linear expression is negative?
>
> This is a good question that should be clarified. We confirm that $\hat D_{ab}$ is a **negative value** in Prop. 1 and list some reasons as follows:
> - Our method is still defined as **likelihood optimization** based on **node similarity**, despite the objective being steered to represent the global shortest path structure. Instead of optimizing the co-occurrence of nodes on **random paths**, we introduce a distance resampling strategy to describe that on **random shortest paths**. It means node pair with **smaller** shortest path distance desires to **larger** score on similarity, which induces a **negative correlation** between such two variables.
> - Our method **performs well** for representing the global shortest path structure, despite node similarity **varying negatively** with the actual shortest distance. According to the proof of Thm. 2, for any $v_a$, $\hat D_{ab} < 0$ is always satisfied when $0<\alpha<1$ and $\log A > 0$​. Thus, $|\hat D_{ab}|$ could be regarded as a decent representation of the shortest distance.
>
> > For proper parameter alpha and sufficiently large distance pair vi and vj, does proposition 1 imply a constant distance distortion for this embedding?
>
> We hold the view that only **limited order proximity** of each node is **well-restricted** in a constant distance distortion after a sufficient number of iterations. Some discussions regarding this are listed as follows:
> - First, there's no doubt that iterative optimization of embeddings is strictly conformed to the objectives defined by Eq. 12 and 13. But one should note that this conclusion is supported mainly by **training samples** of node pairs. Since we adopt a **truncated** random walk in Alg. 1, node pair with **larger** distance **than** walk length is agnostic during optimization.
> - Second, the above property induces **minimal impact** on **long-distance** representation. As is compared with other walk patterns in Appendix A.11, BC-based walk is ensured to cover a wider range of shortest distances **almost proportional** to its **path length** in diverse graph structures. It is implied that for any graph with a **constant** diameter (which is satisfied for plenty of real-world graphs), BC-based walk with **matched path lengths** could provide sufficient prior structural observation for the subsequent optimization.

---

### Official Review · Reviewer_28dr · 2021-11-03

**Correctness:** 3
**Technical Novelty And Significance:** 2
**Empirical Novelty And Significance:** 2
**Recommendation:** 3
**Confidence:** 4

**Main Review:**



- The authors in their abstract state "embedding-based distance prediction has made a breakthrough in both efficiency and accuracy", but the experimental results do not illustrate such a performance, at least for prior methods.

- Continuing on the previous point, Chanpuriya et al. proved in their paper "Node Embeddings and Exact Low-Rank Representations of Complex Networks" that one can obtain low-dimensional embeddings that perfectly encode the graph structure, and thus all-pairwise shortest path distances. While the authors do not correlate the shortest path distances to Euclidean or some other distance in the embedding, it is a worthwhile competitor method as in contrast to node2vec/deepwalk is provably encoding the fully graph structure.

- I would expect to see some toy synthetic experiments that illustrate the power of using BCs over the standard RW based node embedding methods. For instance, it is clear that the proposed RW in the dumbell graph is more likely to explore soon nodes from the second clique.

- Following up on the example, there are plenty of other random walks one can define  that bias the random walk to escape from low conductance sets (e.g., biasing the walk towards nodes that are incident to high effective resistance edges etc.), and thus explore the graph in a more BFS-like way.

- The computational overhead (Time and memory) occurred by the proposed framework is significant. Why would one prefer to use this approach for shortest path distance computation? While it is clear that if one has a good embedding method for downstream machine learning tasks, and this method also can yield accurate shortest path distance estimation (or even in a relative sense, as in the spirit of theorem 2), that is fine. But this is not the case here, or at least it has not been illustrated that the resulting node embeddings are useful for downstream ML tasks.  See for instance, this set of papers related to shortest path distances.

              1. Abraham, I., Delling, D., Fiat, A., Goldberg, A. V., & Werneck, R. F. (2011, July). VC-dimension and shortest path algorithms. In International Colloquium on Automata, Languages, and Programming (pp. 690-699). Springer, Berlin, Heidelberg.
              2. Cohen, E., Halperin, E., Kaplan, H., & Zwick, U. (2003). Reachability and distance queries via 2-hop labels. SIAM Journal on Computing, 32(5), 1338-1355.

- There should be at least a comparison to such combinatorial approaches, or solid experiments that show that embeddings you propose are also useful in other tasks (or ideally both).

- Theorem 1 is not clear. How do you let k tend to infinity? What about graphs with n->inf, but with constant diameter (e.g., dumbell graph). Improving the readability of the theorem would be helpful.

- Finally, the writeup can be improved. E.g., claims as the one in the abstract "It should also outperform existing methods in other graph structure-related applications" should be avoided without concrete proofs or experimental evidence.

- Report total run times for all methods.

**Summary Of The Paper:**

In this work the author(s) proposed a framework for node embeddings that captures better shortest path distances for undirected graphs. Specifically, the authors propose a new random walk framework based on betweenness centralities, and a distance resampling strategy that uses the shortest path distances, and the betweenness centrality scores and is shown to capture well shortest path distances (Proposition 1).  The authors also evaluate their framework experimentally, verifying that compared to other popular node embedding methods, it can be used to represent shortest path distances faithfully.

**Summary Of The Review:**

Based on my comments above, I suggest that the paper is not ready yet for publication, despite having some interesting ideas. I look forward to the authors' addressing my  comments above.

---

> ### Author Response · Authors · 2021-11-16
> **Response to Reviewer 28dr (1/3)**
>
> > The experimental results do not illustrate such a performance that "embedding-based distance prediction has made a breakthrough in both efficiency and accuracy."
>
> Admittedly, this paper aims to improve the performance of embedding-based distance prediction by proposing a much more powerful shortest distance embedding method instead of providing some detailed evaluation between the embedding-based model and other research branches. However, since the significance of embedding-based distance prediction is cited as a background[1-2], we discuss some of the advantages here:
>
> - For efficiency, both embedding-based and landmark-based methods have **constant query time** compared to Oracle-based ones and other exact distance methods. Even in comparison with landmark-based methods, **GPU-only computation** on batches of node embeddings also possesses **low latency** in the real-time response.
> - For accuracy, embedding-based methods show competitive results in mRE. Although some landmark-based models could be on par with such performance, they often **rely heavily on landmark selection** and could **not be simply generalized** on different graph structures.
>
> In conclusion, we think embedding-based distance prediction provides a **decent trade-off** among off-line processing time, storage and online query time.
>
> [1] Z. Xiaohan, A. Sala, C. Wilson, Z. Haitao, and Z. Ben Y. Orion: Shortest path estimation for large social graphs. In Proceedings of the 3rd Wonference on Online Social Networks, WOSN'10, pp. 9, USA, 2010. USENIX Association.
>
> [2] Fatemeh Salehi Rizi, Joerg Schloetterer, and Michael Granitzer. Shortest path distance approximation using deep learning techniques. In 2018 IEEE/ACM International Conference on Advances in Social Networks Analysis and Mining (ASONAM), pp. 1007–1014. IEEE, 2018.
>
> > Comparison with "Node Embeddings and Exact Low-Rank Representations of Complex Networks， Chanpuriya et al."
>
> Thank you for citing such an interesting embedding method and relating it with our work. The comparison between our method and the above work (LPCA([1])) and some analysis are presented in Appendix A.2.
>
> We consider LPCA proposes an exciting perspective to embed complex sparse graphs perfectly into low-rank representations, which is helpful for downstream ML tasks. Nevertheless, we need to address **the limitation of LPCA** in shortest distance representation:
> - First and most importantly, it should be clarified that a perfect representation of graph structure is **not equal to** that of graph shortest path structure, since they have a large "calculation gap". According to the Floyd-Warshall algorithm([2]), even if each node is aware of all related path structures, inference of the exact shortest path structure also needs up to $O(N^3)$ complexity.
> - Second, LPCA shows **poor embedding performance** on some **relatively-dense graphs**, despite converging fast on some sparse graphs. As illustrated in Fig. 5 and Tab. 4, LPCA meets a bottleneck on the Facebook graph and consumes a **long time to converge**.
>
> Compared to the above method, the motivation of this paper is to directly embed graph shortest distance matrix with sub-linear time complexity for **fast and accurate** online queries of shortest distance. To the best of our knowledge, there is **no existing embedding method** that could **directly and perfectly represent shortest path structures** in linear time.
>
> Moreover, our method takes a more flexible objective to represent shortest distances, since only high-level restrictions are implicitly exerted on the embedding space by Eq. 12 and Eq. 13.
>
> [1] Sudhanshu Chanpuriya, Cameron Musco, Konstantinos Sotiropoulos, and Charalampos Tsourakakis. Node embeddings and exact low-rank representations of complex networks. Advances in Neural Information Processing Systems, 33, 2020.
> [2] Robert W Floyd. Algorithm 97: shortest path. Communications of the ACM, 5(6):345, 1962.

---

> > ### Author Response · Authors · 2021-11-16
> > **Response to Reviewer 28dr (2/3)**
> >
> > > Some toy synthetic experiments that illustrate the power of using BCs over the standard RW based node embedding methods.
> >
> > We adopt the above reasonable and valuable suggestions and test BC-based walk on **six synthetic graph datasets of divergent topology** to solid our conclusion.  The visualization of each graph and comparison results with other walking patterns are illustrated in Appendix A.7.5 and A.11, respectively.
> > - For circle graphs and tri-circle graphs, BC-based walk tends to choose the exit nodes of each circle since they share a large BC gain by locating on the shortest path between inner and outer nodes.
> > - For triangle graphs, transitions on every triangle clique tend to move forward since the number of nodes beyond the current clique is often larger than that of inner nodes, contributing to more shortest paths.
> > - For tree graphs, each transition appears to move forward from the root to the leaves, and all walking patterns show a similar exploration.
> > - For spiral graphs, although some exit nodes are located on the circle, BC gains on exit nodes and inner nodes on circles are usually on par, which misleads the direction choice of the next transition. Thus, BC-based walk only performs slightly better than others.
> > - For net graphs, all walking patterns are limited on exploration distance, since BC values on different nodes do not change considerably.
> >
> > To conclude, the above results show that BC-based walk possesses a much better all-around performance to capture remote nodes with large shortest distances.
> >
> > > There are plenty of other random walks one can define that bias the random walk to escape from low conductance sets (e.g., biasing the walk towards nodes that are incident to high effective resistance edges etc.), and thus explore the graph in a more BFS-like way.
> >
> > We define a BFS-like random walk by considering a significant probability (with $p \approx 0.901$) to move out from the current clique. The experimental results are also available in Appendix A.11.
> >
> > The results show in circle, triangle and tri-circle graphs that the BFS-like walking pattern has a slight edge over others, but is not as competitive as BC-based random walk, mainly because of the loss of global structure to the root of each walking path.
> >
> > > Theorem 1 is not clear. How do you let k tend to infinity?
> >
> > Thank you for pointing out this issue. We admit the condition in Thm. 1 is inapposite, since the order of neighborhoods on any node should be limited with respect to the graph's diameter. If we take the suppose of sparse graph, the above condition could be rewritten as $h-k + 1 \rightarrow \mathcal E(v_a)$​​ or $k \rightarrow \mathcal E(v_a) - 1 - h$​​​, where $\mathcal E(v_a)$ is the eccentricity of $v_a$ and $h$ is the order of neighborhoods. The corresponding write-up in the paper has been revised in the updated manuscript.

---

> > > ### Author Response · Authors · 2021-11-16
> > > **Response to Reviewer 28dr (3/3)**
> > >
> > > > There should be at least a comparison to such combinatorial approaches, or solid experiments that show that embeddings you propose are also useful in other tasks (or ideally both).
> > >
> > > We hold the view that comparisons with other branches of distance prediction (such as the landmark-based or Oracle-based) seem to be less meaningful for two reasons.
> > >
> > > - First, **no inter-branch de facto standard** is proposed for shortest distance prediction at present. Thus, it might be difficult to determine parameter setup for embedding-based and landmark-based models, which is essential for convincible comparisons.
> > > - Second, the latest work DADL([1]) has **already highlighted** the practicality and efficiency of **GRL techniques** in **shortest distance estimation**, which motivates us to improve the performance of current embedding methods on shortest path representation. Thus, the main topic of this paper should be **concentrated on the representation** itself (including comparisons in complexity and accuracy) **instead of comparing** it with **non-representation-based distance estimation**. In the previous response, we also point out that **embedding-based** distance prediction framework is **indispensable for some applications** due to its **unique trade-off** among off-line processing time, storage and online query time.
> > >
> > > Furthermore, we locate the motivation of this paper currently at improving the accuracy performance of embedding-based distance prediction by proposing a much more powerful shortest distance embedding method, despite some of the results appearing to be helpful for downstream ML tasks. Several corresponding experimental results support our conclusions in **both accuracy and complexity**.
> > >
> > > - For accuracy, it is evaluated that in the research branch of embedding-based distance predictions, BCDR **outperforms** the latest work DADL that utilizes general GRL techniques for node embeddings. Moreover, there are several pieces of evidence(in Tab. 3, Tab. 4, Tab. 5 and Fig. 9) implying that our method also shows **much better performance than other existing GRL methods** with respect to the representation of graph's shortest path structure.
> > >
> > > - For complexity, our method is evaluated to possess **competitive processing time** for **both sparse graphs and relatively-dense graphs**.  It is illustrated in Fig. 10 and Fig. 11 that BCDR possesses a similar processing time with general random walk-based embedding in sparse graphs via the adjustment of the sampling coefficient $\beta$. Moreover, there are some pieces of evidence in Tab. 4 that reveal our method performs faster than LPCA([2]) on relatively-dense graphs.
> > >
> > > [1] Fatemeh Salehi Rizi, Joerg Schloetterer, and Michael Granitzer. Shortest path distance approximation using deep learning techniques. In 2018 IEEE/ACM International Conference on Advances in Social Networks Analysis and Mining (ASONAM), pp. 1007–1014. IEEE, 2018.
> > >
> > > [2] Sudhanshu Chanpuriya, Cameron Musco, Konstantinos Sotiropoulos, and Charalampos Tsourakakis. Node embeddings and exact low-rank representations of complex networks. Advances in Neural Information Processing Systems, 33, 2020.
> > >
> > > > Claims as the one in the abstract "It should also outperform existing methods in other graph structure-related applications" should be avoided without concrete proofs or experimental evidence.
> > >
> > > We admit this assumption is inapposite here and omit it in the updated manuscript.
> > >
> > > > The computational overhead (Time and memory) occurred by the proposed framework is significant. Report total run times for all methods.
> > >
> > > We admit the concerns on processing time should be addressed and provide the run time comparisons between our method and **general random walk** on the three real-world datasets in Appendix A.10. As discussed in Sec. 3.3 and Sec. 4.2, choices of **output walk length** and the **number of output walks** largely influence processing time. The experimental results reveal that the total run time of our method could be **reduced to keep pace with general random walk** in sparse graphs, though for relatively dense graphs, there is some room for further optimization.

---

> > > > ### Comment · Reviewer_28dr · 2021-11-18
> > > > **additional question**
> > > >
> > > > I appreciate a lot the detailed comments. Are you aware of any paper that illustrates that estimating shortest path distances from node embeddings can outperform the state-of-the-art combinatorial approaches I mentioned? I took a look at the papers you suggested, and even if they cite some combinatorial approaches (e.g., landmark based) they do not compare against the known state-of-the-art. For instance the DADL method you cite, does not contain such a comparison.

---

> > > > > ### Author Response · Authors · 2021-11-19
> > > > > **Response to Reviewer 28dr**
> > > > >
> > > > > Thanks for your timely response.
> > > > >
> > > > > We would like to deliver some relevant discussions from previous work between node-embedding-based distance estimation and others, despite there being **little** comparison **directly with** the methods mentioned in [1] and [2].
> > > > >
> > > > > We consider [2] initially introduces a compressing strategy to shortest distance query problems by 2-hop label on each node, and shows an efficient algorithm to find the almost optimal solution with $O(nm^{\frac{1}{2}})$ space complexity. For any node $v_a$ and $v_b$, as long as the node label sets $L(v_a)$ and $L(v_b)$ contains some node $v_x$ just on one of the shortest paths of $v_a,v_b$, the shortest distance $D_{ab}$ could be found **exactly**. But the above method suffers from the cost of traversal on graphs while constructing such a data structure, and possesses $O(n^{\frac{3}{2}})$ space complexity on sparse graphs, even worse for dense graphs. Improvement([3-4]) addressing this is to heuristically define a set of $v_x$ as the path anchors, which appears to be a landmark-based approach.
> > > > >
> > > > > Then, in an early work([5]) of embedding-based distance estimation before DADL([6]), an intuitive strategy for **predicting shortest paths** from mutual shortest distances is proposed, and some **comparisons with the landmark-based approaches** (include the state-of-art in that work) are illustrated. The experimental results shown in Fig. 10-12 and Table VI of [5] reflect that the embedding-based approach **strikingly outperforms** others([3-4]) in **response time** with **competitive accuracy**.
> > > > >
> > > > > [1] Abraham, I., Delling, D., Fiat, A., Goldberg, A. V., & Werneck, R. F. (2011, July). VC-dimension and shortest path algorithms. In International Colloquium on Automata, Languages, and Programming (pp. 690-699). Springer, Berlin, Heidelberg.
> > > > >
> > > > > [2] Cohen, E., Halperin, E., Kaplan, H., & Zwick, U. (2003). Reachability and distance queries via 2-hop labels. SIAM Journal on Computing, 32(5), 1338-1355.
> > > > >
> > > > > [3] Das Sarma A, Gollapudi S, Najork M, et al. A sketch-based distance oracle for web-scale graphs[C]//Proceedings of the third ACM international conference on Web search and data mining. 2010: 401-410.
> > > > >
> > > > > [4] Gubichev A, Bedathur S, Seufert S, et al. Fast and accurate estimation of shortest paths in large graphs[C]//Proceedings of the 19th ACM international conference on Information and knowledge management. 2010: 499-508.
> > > > >
> > > > > [5] Zhao X, Sala A, Zheng H, et al. Fast and scalable analysis of massive social graphs[J]. arXiv preprint arXiv:1107.5114, 2011.
> > > > >
> > > > > [6] Fatemeh Salehi Rizi, Joerg Schloetterer, and Michael Granitzer. Shortest path distance approximation using deep learning techniques. In 2018 IEEE/ACM International Conference on Advances in Social Networks Analysis and Mining (ASONAM), pp. 1007–1014. IEEE, 2018.

---

### Official Review · Reviewer_FZhh · 2021-11-07

**Correctness:** 3
**Technical Novelty And Significance:** 3
**Empirical Novelty And Significance:** 2
**Recommendation:** 3
**Confidence:** 4

**Details Of Ethics Concerns:**

No concern at this time.

**Main Review:**

The paper presents an interesting approach to a difficult problem, yet also presents opportunities for improvement in terms of motivation, breadth of experimental comparison, and scope of experiments.

The motivation argument appears to be false: it is claimed that one should take into account remote nodes under a graph’s shortest distance metric, yet it is not clear why such remote nodes should be relevant. It is argued that PMI optimization should be avoided, as that reflects local similarity rather than distance. It is shown that this approach can accommodate an optimization objective of global distance matrix reconstruction.

The method is compared experimentally to classic graph embedding methods on several real world datasets. Out of the methods compared to, the most recent one appears to be DADL. While the work of Shaosheng et al., GraRep, is referenced several times, that is not used in the experimental comparison. Besides, most recent advances in graph representation learning, including methods such as NetMF, NetSMF, VERSE, FREDE, and ProNE, are not included in the comparison. A vague claim is made that the proposed method should outperform existing ones in other application, yet no evidence is provided.

The largest graph use in the comparison comprises some 5K nodes. There is no suggestion that the proposed embedding would scale to graph sizes typically appearing in graph embedding applications. The complexity analysis appearing in Table 1 appears to reflect a sparse graph, as discussion in Section A.5. That complexity would allow running the algorithm on larger data, yet no experiment to that effect is shown.

**Summary Of The Paper:**

This paper proposes a method to construct graph embeddings that are well-tuned for answering shortest-distance queries (SDQs), based on betweennes centrality distance sampling. The main idea appears to be that the betweenness centrality measures helps identify nodes that could serve as landmarks for distance calculations, hence an distance-oriented embedding anchored on betweenness centrality is bound to perform well. Distance are resampled from walk paths, and a step of other methods based on pointwise mutual information (PMI) optimization is abandoned.

**Summary Of The Review:**

The paper presents an interesting approach to a difficult problem, yet also presents opportunities for improvement in terms of motivation, breadth of experimental comparison, and scope of experiments.

---

> ### Author Response · Authors · 2021-11-16
> **Response to Reviewer FZhh (1/2)**
>
> > It is claimed that one should take into account remote nodes under a graph's shortest distance metric, yet it is not clear why such remote nodes should be relevant.
>
> It is necessary to point out that different from other graph-based tasks, inference of shortest path distance requires the awareness of global path structure. It means embedding techniques relying only on nodes' local features is insufficient to solve such a problem, while considering the correlation with remote nodes is helpful.
>
> We list as follows the detailed discussion about our method and other GRL techniques to clarify the motivation of this paper.
>
> - For common node-level tasks (e.g., node classification and link prediction), first- and second-order proximity seem to be enough for predicting edges or node labels, since the marginal distribution of such information could be inferred locally according to D-separation in a probabilistic graphical model. GRL techniques towards these tasks mainly describe local similarity over nodes' neighborhoods, with **little attention to representing high-order proximity**. But things are different here. As presented in [1], the shortest distance is inferred only by embeddings of source and destination nodes. The solution to this problem appears to be difficult if both the embeddings are not directly consistent with the global path structure. In detail, for node $v_b$ and $v_c$ away from $v_a$ with different shortest distances $D_{ab}$ and $D_{ac}$, the results are fairly intractable since both $v_b$  and $v_c$ share little similarity with $v_a$. That explains why we need to embed the correlation with remote nodes and consider different order proximity of equivalent importance.
> - How to embed the correlation with remote nodes? We think the most efficient way is to utilize random walk strategies, which possess linear complexity and a large receptive field. Moreover, the intuitive ideas appear to be:
>     - **Sufficiently high-order proximity** should be **embedded** on each node.
>     - **Different order proximity** should be **discerned** well with respect to the measure of shortest distance.
> - The above two aspects are addressed by **BC-based random walk** and **Distance Resampling strategy** in Sec. 3.1 and Sec. 3.2, respectively.
>
> [1] Fatemeh Salehi Rizi, Joerg Schloetterer, and Michael Granitzer. Shortest path distance approximation using deep learning techniques. In 2018 IEEE/ACM International Conference on Advances in Social Networks Analysis and Mining (ASONAM), pp. 1007–1014. IEEE, 2018.
>
> > It is argued that PMI optimization should be avoided, as that reflects local similarity rather than distance. It is shown that this approach can accommodate an optimization objective of global distance matrix reconstruction.
>
> We clarify the connection between our method and PMI-based optimization to answer this question.
>
> Note that PMI-based optimization in graphs essentially reflects node-level similarity under some structural bias implied by observation.
>
> - For general random walk methods, this observation is the co-occurrence of node pairs on any **random paths**, which does not reflect shortest distances according to Eq. 3. Therefore, similarities measured by PMI  have no direct relationship with the shortest distance between nodes.
> - For our method, this observation is the co-occurrence of node pairs on any **random shortest paths**. As shortest path problems have optimal substructure, the subpath between two nodes on one shortest path is also the shortest path between them. Under this condition, PMI similarities just reflect the global shortest distance matrix.
> - But how to find that shortest path structure in a graph? Instead of directly figuring them out by shortest path algorithms, we adopt a flexible resampling process from random paths to simulate random shortest paths with linear time complexity.
>
> To conclude, our method could be interpreted as a variant of PMI-based optimization under the graph's shortest path structure.

---

> > ### Author Response · Authors · 2021-11-16
> > **Response to Reviewer FZhh (2/2)**
> >
> > > GraRep and most recent advances in GRL,  including methods such as NetMF, NetSMF, VERSE, FREDE, and ProNE, are not included in the comparison.
> >
> > We agree with the reviews and provide some further comparisons with GraRep ([1]) and some of the most recent models in Appendix A.2 and A.9. Thereinto, as the source code of FREDE and ProNE is not available at present, comparison with them will be appended in the future. As an alternative, a latest work LPCA ([2]) is included, which perfectly encodes the graph's global structure into low-dimensional representations. The experimental results show our method also outperforms them in shortest distance estimation.
> >
> > As discussed in the previous response, we consider the majority of general GRL techniques endeavors to figure out the most similar nodes in limited neighborhoods without discerning an elaborate hierarchy of different order proximity. As a result, most of them are inefficient in embedding shortest path structures, especially for remote nodes, as illustrated in Tab. 5 and Fig. 9.
> >
> > [1] C. Shaosheng, L. Wei, and X. Qiongkai. Grarep: Learning graph representations with global structural information. In Proceedings of the 24th ACM international on conference on information and knowledge management, pp. 891–900, 2015.
> >
> > [2] Sudhanshu Chanpuriya, Cameron Musco, Konstantinos Sotiropoulos, and Charalampos Tsourakakis. Node embeddings and exact low-rank representations of complex networks. Advances in Neural Information Processing Systems, 33, 2020.
> >
> > > A vague claim is made that the proposed method should outperform existing ones in other application, yet no evidence is provided.
> >
> > We admit this assumption is inapposite here and omit it in the updated manuscript.
> >
> > > There is no suggestion that the proposed embedding would scale to graph sizes typically appearing in graph embedding applications.
> >
> > We answer this question from two aspects of complexity and accuracy.
> >
> > - For complexity, as discussed in Table 1 and Appendix A.5, our method shows a **sub-linear off-line time complexity** for **sparse graphs** (e.g., Cora and GrQc datasets), and a little bit of time penalty for relatively-dense graphs(e.g., Facebook dataset). Besides, the time complexity could be further reduced by adjusting sampling parameters $w_{out}, l_{out}$. Detailed processing time results are provided in Fig. 10 and show our method is on par with the general random walk model concerning sparse graphs. Even for a relatively-dense graph, Tab. 4 also reveals that our method performs better than the latest embedding model LPCA ([1]) in processing time.
> > - For accuracy, we consider the **complexity of the graph's structure** rather than **graph size** significantly influences shortest distance estimation, and test BC-based walk with other random walking patterns on **six synthetic graph datasets of divergent topology**. The properties and visualization of these datasets are available in Appendix A.7.5. The results in Appendix A.11 show that our method possesses a better all-around performance to capture remote nodes with large shortest distances.
> >
> > Admittedly, there are also plenty of large graph datasets available to test the performance of our method for both dense and large graphs, such as BlogCatalog and Twitter. And the additional comparison on these datasets will be later appended in this paper.
> >
> > [1] Sudhanshu Chanpuriya, Cameron Musco, Konstantinos Sotiropoulos, and Charalampos Tsourakakis. Node embeddings and exact low-rank representations of complex networks. Advances in Neural Information Processing Systems, 33, 2020.

---

### Author Response · Authors · 2021-11-16
**General Response**

We would like to highly appreciate all reviewers for their constructive feedback and comments. Several experimental results and other revisions are included in the updated manuscript, as are listed as follows.
- Connection to the latest graph structure decomposition method(LPCA,[1]) is discussed for both **motivation and empirical performance** in Appendix A.2.
- A **broadened comparison** with several of the latest graph embedding methods is provided in Appendix A.9. Furthermore, the length-level accuracy of each method is also illustrated in Fig. 8 and Fig. 9.
- Six synthetic graph datasets of diverse shortest path structures are available in Appendix A.7.5 to **enlarge the scope** of experiments.
- Further comparison on diverse graph structures of walking patterns is presented in Appendix. A.11 to address some concerns about **cycle-related** limitations.
- Further comparison on **run time** is available in Appendix. A.10.
- Some trivial improvement on readability.

Additionally, the motivation of this paper is clarified with respect to each concern from the reviewers, respectively, and we look forward to further discussion on this topic.

[1] Sudhanshu Chanpuriya, Cameron Musco, Konstantinos Sotiropoulos, and Charalampos Tsourakakis. Node embeddings and exact low-rank representations of complex networks. Advances in Neural Information Processing Systems, 33, 2020.

---

### Decision · Program_Chairs · 2022-01-20

**Decision:**

Reject

**Comment:**

The authors propose a new methods for graph shortest distance embedding method called BCDR based on betweenness centrality. Then they show that the method is competitive both theoretically than experimentally with existing work.

After a discussion with the reviewers and after considering the nice changes in the paper and explanation in the rebuttal we agree that the paper contains some very interesting ideas but it is not probably ready for publication. The comparison with previous works is, in fact, still a bit limited and it should be extended. In addition the algorithm should also be tested on larger datasets.